# MobileODE: An Extra Lightweight Network

**Le Yu[1], Jun Wu[1], Bo Gou[1], Xiangde Min[2], Lei Zhang[1], Zhang Yi[1], Tao He[1]***

[1]Machine Intelligence Laboratory, Sichuan University
[2]Tongji Medical College, Huazhong University of Science and Technology
[1]{yule,junwu,goubo7795}@stu.scu.edu.cn [1]{leizhang,zhangyi,tao_he}@scu.edu.cn
[2]mxianade@tih,timu.edu.cn

## Abstract

Depthwise-separable convolution has emerged as a significant milestone in the lightweight development of Convolutional Neural Networks (CNNs) over the past decade. This technique consists of two key components: depthwise convolution, which captures spatial information, and pointwise convolution, which enhances channel interactions. In this paper, we propose a novel method to lightweight CNNs through the discretization of Ordinary Differential Equations (ODEs). Specifically, we optimize depthwise-separable convolution by replacing the pointwise convolution with a discrete ODE module, termed the ***Channelwise ODE Solver (COS)***. The COS module is constructed by a simple yet efficient direct differentiation Euler algorithm, using learnable increment parameters. This replacement reduces parameters by over $98.36\%$ compared to conventional pointwise convolution. By integrating COS into MobileNet, we develop a new extra lightweight network called MobileODE. With carefully designed basic and inverse residual blocks, the resulting MobileODEV1 and MobileODEV2 reduce channel interaction parameters by $71.0\%$ and $69.2\%$, respectively, compared to MobileNetV1, while achieving higher accuracy across various tasks, including image classification, object detection, and semantic segmentation. The code is available at https://github.com/cashily/MobileODE.

## 1 Introduction

The design of lightweight networks has become a central focus in computer vision, aiming to balance high performance with low computational costs. This is especially crucial for resource-constrained devices like mobile devices, embedded systems, and edge computing platforms, which have limited computational power and storage capacity. As these devices face growing demands for efficient model deployment, recent advancements in lightweight Convolutional Neural Networks (CNNs) have led to the development of efficient architectures that provide practical solutions to these challenges. Currently, the development of lightweight networks has largely focused on optimizing the MobileNet series and integrating self-attention mechanisms. MobileNetV1 [Howard, 2017] was the first to introduce a lightweight network by employing depthwise-separable convolutions, marking a milestone in the lightweight evolution of CNNs over the past decade. MobileNetV2-V4 [Sandler et al., 2018, Howard et al., 2019, Qin et al., 2025] enhanced MobileNetV1 by incorporating inverted residuals and linear bottlenecks, network architecture search, and universal inverted bottlenecks, respectively. GhostNet [Han et al., 2020], ShuffleNet [Zhang et al., 2018], and MobileOne [Vasu et al., 2023b] optimized depthwise-separable convolutions while maintaining the core design of separating spatial and channel computations. More recently, self-attention mechanisms have been introduced to further improve MobileNet variants by enhancing global context. Models like FastViT [Vasu et al., 2023a], MobileViT [Mehta and Rastegari, 2021], EfficientFormerV2 [Li et al., 2023], and RepViT [Wang et al.,

---

*Corresponding author

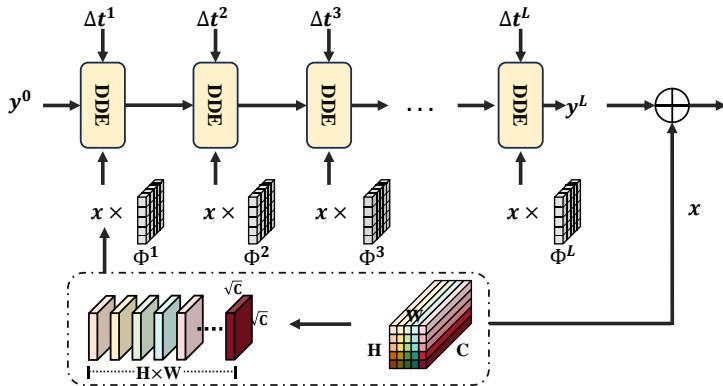

Figure 1: Detailed Structure of the COS Module. We propose the Direct Differentiation Euler (DDE) algorithm as a highly efficient method for approximating solutions in high-dimensional spaces. Building on DDE, we develop the Channelwise ODE Solver (COS) module, which enhances interchannel interactions by mapping the input to its global attractor.

2024a] have transformed MobileNet-like architectures into Vision Transformers (ViTs), significantly boosting their performance. However, their parameter counts still remain comparable to, or even exceed, the original designs.

The primary mechanism behind depthwise-separable convolution lies in the separation of spatial and channel computations, where depthwise convolution is responsible for extracting spatial information and pointwise convolution enhances channel interaction. However, this separable design reduces the receptive field of channelwise computations due to the reduction in the number of parameters. Specifically, the $1 \times 1$ convolution only performs a linear mapping on features in the form of $1 \times 1 \times C$, where $C$ represents the number of channels. Moreover, the computational cost of spatial operations is lower than that of channelwise computations. To address this, we aim to enhance the global receptive field of channel computations by adopting a non-linear mapping approach, thereby introducing an extra lightweight network with parameters comparable to those used in spatial computations.

Recently, Neural Memory Ordinary Differential Equations (nmODEs) [Yi, 2023] have shown significant advancements in efficiently modeling non-linear mappings. This is primarily due to their unique characteristic of having a single global attractor, which implies that all trajectories within the dynamical system defined by nmODEs will ultimately converge to this attractor [Wills et al., 2005]. This property enables nmODEs to establish a robust connection between the input space and the memory space [Poucet and Save, 2005]. These advancements have been particularly impactful in fields of computer vision, such as medical image segmentation [He et al., 2024, 2023], video captioning [Artham and Shaikh, 2024], and image recognition [Niu et al., 2024, Luo et al., 2024]. The discrete nmODEs have been employed to design lightweight network architectures. [He et al., 2024] introduced various techniques, including Euler's method, Heun's method, and linear multi-step method, to reduce the complexity of the decoder in U-like networks. These methods have been demonstrated effective when integrated into state space models [Wang et al., 2024b]. While discrete nmODEs have streamlined network architectures for semantic segmentation tasks, there remains a need for research focused on developing generalized network structures applicable to diverse tasks.

To the best of our knowledge, this paper is the first work to develop the discrete nmODEs to construct an extra lightweight MobileNet-like network. The main contributions of this paper are as follows:

- We propose a discrete nmODEs module, named Channelwise ODE Solver (COS), illustrated in Fig. 1, which is structured by a novel Direct Differentiation Euler (DDE) algorithm. The DDE enables the time increments learnable during network training, verifying the gradient descent stability of the COS module.

- We build a novel extra lightweight network, named MobileODE, which incorporates the COS modules. Both the basic block (COS-base) and the inverse residual block (COS-inv) perform efficiently. The resulting MobileODEV1 and MobileODEV2 reduce the parameters for channel interaction by $71.0\%$ and $69.2\%$, respectively, compared to MobileNetV1, as shown in Tab. 1.

- Extensive experiments across various tasks, including image classification, object detection, and semantic segmentation, demonstrate the effectiveness of the proposed MobileODEV1 and MobileODEV2 architectures.

## 2  Related work

**MobileNet Series**  MobileNet [Howard, 2017] introduced the classic depthwise-separable convolution, significantly reducing the model's parameter count and computational complexity. Since then, numerous improvements based on this module have emerged. MobileNetV2 [Sandler et al., 2018] enhanced feature dimensions by adding a pointwise convolution before the depthwise-separable convolution and introduced the linear bottleneck to stabilize network performance. MobileNetV3 [Howard et al., 2019] updated the inverted bottleneck block from MobileNetV2 by incorporating the Squeeze-and-Excitation (SE) module [Hu et al., 2018] and the Hard-Swish activation function, as well as introducing Neural Architecture Search (NAS) to automatically discover optimal network architectures. MobileNext [Zhou et al., 2020] introduced the SandGlass module, which establishes shortcuts based on high-dimensional features. MobileNetV4 [Qin et al., 2025] advanced the series further by introducing the Universal Inverted Bottleneck (UIB) block, which incorporates depthwise convolution before the inverted bottleneck module. MobileOne [Vasu et al., 2023b] returned to the original separable convolution, introducing reparameterization to merge multiple branches into a single-branch structure. Despite these several iterations of improvement, the fundamental nature of depthwise-separable convolution remains unchanged: it uses depthwise convolution to extract spatial information and pointwise convolution to enhance channel interaction. This paper optimizes the channel interaction of depthwise-separable convolution through a discrete nmODEs method.

**MobileNets with Self-Attentions**  The integration of Transformer architectures with depthwise-separable convolutional blocks to better learn global representations has become a major research focus. FastViT [Vasu et al., 2023a], a hybrid vision transformer architecture, placed the computationally expensive self-attention mechanism in the later stages of the network, where the resolution is lower. It also used structural reparameterization to reduce memory access costs and improve network accuracy. MobileViT [Mehta and Rastegari, 2021] seamlessly combined the ViT structure with convolutions by introducing the MobileViT block, which replaced some of the inverted bottleneck blocks in the MobileNetV2 model, achieving global receptive fields at a lower computational cost. EfficientFormerV2 [Li et al., 2023] proposed a fine-grained joint search strategy that simultaneously optimized latency and parameter count to find efficient architectures, enabling high-speed execution of ViT. RepViT [Wang et al., 2024a] improved upon the MobileNetV3 block by separating token mixers and channel mixers, employing structural reparameterization techniques to enhance learning. These models achieved a balance between speed and accuracy by lightweighting the ViT architecture to avoid the expensive computational cost.

**Lightweight Discrete ODE Models**  Ordinary Differential Equations (ODEs) provides a conceptual framework for neural networks and has been widely applied in mathematics and physics [Chen et al., 2018, Shou et al., 2024]. Neural memory Ordinary Differential Equations (nmODEs) [Yi, 2023] was designed to fully utilize the memory capacity provided by dynamic systems. nmODEs enhanced the non-linear representation of neural networks through implicit mapping and non-linear activation functions. Discrete nmODEs has demonstrated its versatility and effectiveness in treamlining network structures. For instance, He et al. [2024] employed explicit Euler's method, Heun's method, and linear multi-step method for discretization and further introduced three plug-and-play decoders that reduce the number of parameters and FLOPs while maintaining performance when embedded into various U-shaped networks. Similarly, He et al. [2023] followed the explicit Euler's method and applied it to liver tumor segmentation. Wang et al. [2024b] leveraged the powerful non-linear representation capability of nmODEs combined with state space models, incorporating residual connections to build a medical image segmentation network, called nmSSM-UNet. Additionally, Niu et al. [2024] utilized the discrete nmODEs as the inverse path in a bi-directional network structure, and evaluated the possibility of the bi-directional feedforward network architecture. These methods demonstrate that nmODEs possess strong learning capabilities across various tasks while facilitating lightweight networks, leveraging their inherent advantages in non-linear mappings.

# 3 Method

## 3.1 Direct Differentiation Euler Algorithm

The basic method of discretizing nmODEs is to split the time step $t$ into smaller intervals. The Euler method [Euler, 1845] is one of the simplest and most widely used techniques for solving ODEs numerically. Given an initial state $y(t_0)$ and the basic nmODEs:

$$\dot{y}(t) = -y(t) + f(y(t) + g(x(t), \theta_t)), \tag{1}$$

where, $y(t)$ denotes the internal state, $x(t)$ is the external input, $g(\cdot)$ denotes a linear mapping of $x$, and the $f(\cdot)$ is the non-linear activation function. The discretization of the nmODEs is as follows:

$$y(t + \Delta t) = y(t) + \Delta t \cdot \dot{y}(t), \tag{2}$$

where, $\Delta t$ represents the time increment, and $y(t + \Delta t)$ is an approximate solution at time $t + \Delta t$. In previous related work [He et al., 2023, Wang et al., 2024b, Niu et al., 2024, He et al., 2024], the explicit Euler method is employed for discrete nmODEs to integrate information with minimal parameters, where $\Delta t$ is typically initialized as a fixed constant. Combine Eq. (1) and (2), for each layer $l < L$, we can derive that

$$y^{l+1} = (1 - \Delta t) \cdot y^l + \Delta t \cdot f(y^l + g(x^l, \theta^l)). \tag{3}$$

He et al. [2024] and Niu et al. [2024] initialized $\Delta t$ as $\frac{1}{L}$, where $L$ is the number of discretization layers. However, every ODE solver's discretization introduces approximation errors, which increase with larger time increments. Thus, using a constant to initialize $\Delta t$ can harm the dynamics of nmODEs. This approach not only limits accuracy but also risks making the system uncontrollable.

To address this limitation, we propose the Direct Differentiation Euler (DDE) algorithm, which introduces learnable time increments $\Delta t$ to guide the model's computation, where $\Delta t = \{\Delta t^1, \Delta t^2, \ldots, \Delta t^L\}$, initialized using a Normal distribution. These increments are then passed through a ReLU6 activation function to ensure they remain non-negative and to limit their output range. For each layer $l < L$, we obtain:

$$\begin{cases} \dot{y}^l(t) = -y^l + f(y^l + g(x^l, \theta^l)), \\ y^{l+1} = y^l + \Delta t^l \cdot \dot{y}^l(t). \end{cases} \tag{4}$$

Instead of relying on a fixed $\Delta t$, the DDE method dynamically adjusts $\Delta t$ during training through gradient descent. This allows the model to refine its temporal resolution adaptively, enabling the effects of time increments to accumulate across multiple layers.

The DDE method differs from previous approaches, as shown in Fig. 2. First, prior methods use Eq. (3) as a gating operation that regulates contributions from previous and current layers. In contrast, DDE computes specific partial derivatives layer by layer, solving them directly as intermediate variables to reduce errors. Second, DDE employs sequenced time increments for layer-wise adjustments at each step. In steeper gradient regions, $\Delta t^l$ learns smaller values, while in smoother areas, it can learn larger increments, as shown in Appendix 6.3. Previous gating methods are fixed and lack this flexibility. In Sec. 4.4, we investigate how $\Delta t$

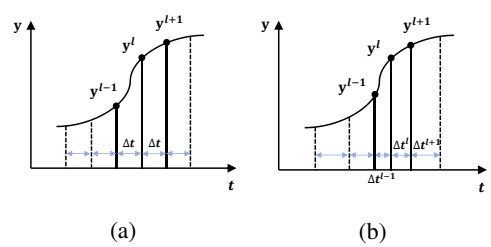

(a)           (b)

Figure 2: The distinction between the explicit Euler algorithm (a) and the proposed DDE algorithm (b).

affects the learning process. We find that as $L$ increases, a fixed $\Delta t$ can impair accuracy, whereas DDE consistently enhances it. Additionally, with deeper layers, the accuracy of features from each layer's outputs progressively improves, demonstrating strong learning capability.

## 3.2 Channelwise ODE Solver

Inspired by the unique global attractor of nmODEs, which allows every input to converge to a unified memory representation, we leverage this characteristic to introduce the COS module. Based on the

DDE algorithm, COS is a lightweight inter-channel computation technique that integrates features with minimal computational cost. It decomposes inter-channel computation into two key steps: inter-channel linear transformation and unified channelwise mapping using DDE.

**Inter-channel linear transformation.** We define a feature tensor $\boldsymbol{x} \in \mathbb{R}^{C \times H \times W}$, where $C$ represents the number of channels, and $H$ and $W$ denote the height and width of the input feature map, respectively. As illustrated in Fig. 1, we perform an inter-channel linear transformation to facilitate channel interaction at each pixel of the feature map, which is formulated as follows:

$$g(\boldsymbol{x}_{\langle i,j \rangle}) = \boldsymbol{x}_{\langle i,j \rangle} \cdot \boldsymbol{\Phi}, \tag{5}$$

where, $\boldsymbol{\Phi} \in \mathbb{R}^{\sqrt{C} \times \sqrt{C}}$ is a learnable matrix, $1 \leq i \leq H, 1 \leq j \leq W$, and $\boldsymbol{x}_{\langle i,j \rangle} \in \mathbb{R}^{\sqrt{C} \times \sqrt{C} \times 1 \times 1}$. Traditional convolutions exhibit quadratic complexity, while our method utilizes a compact $\boldsymbol{\Phi}$ matrix to work in tandem with the learnable time increments. This is the direct reason for the significant reduction in computational cost, transforming a quadratic complexity into linear complexity.

---

**Algorithm 1** Channelwise ODE Solver (COS) with DDE

---

1: **Input:** The feature tensor $\boldsymbol{x}$, learnable weight matrices $\boldsymbol{\Phi} \in \mathbb{R}^{\sqrt{C} \times \sqrt{C} \times L}$, learnable time increments $\Delta \boldsymbol{t} \in \mathbb{R}^L$, initial internal state $\boldsymbol{y}^0 \in \mathbb{R}^{C \times H \times W}$.
2: **Output:** Final output $\boldsymbol{o}$.
3: **for** layer $l = 0$ **to** $L - 1$ **do**
4:     **for** $i = 1, j = 1$ **to** $H, W$ **do**
5:         Calculate the linear transformation $g(\boldsymbol{x}_{\langle i,j \rangle})$ as described in Eq. (5).
6:         Update the internal state $\boldsymbol{y}^{l+1}_{\langle i,j \rangle}$, as shown in Eq. (4).
7:     **end for**
8:     Combine these internal states along the feature map to obtain $\boldsymbol{y}^{l+1}$.
9: **end for**
10: **Reture:** $\boldsymbol{o} = \boldsymbol{y}^L + \boldsymbol{x}$.

---

**Unified channelwise mapping using DDE.** After obtaining the linear transformation $g(\boldsymbol{x}^l)$ of the nmODEs, we employ the DDE algorithm to build the COS internal structure. For an $L$-layer DDE, we first compute the partial derivatives directly and then use the learnable $\Delta t^l$ to calculate the final output of the channelwise mapping, as shown in Eq. (4). The DDE harnesses the advantages of nmODEs by mapping the input to its global attractor, linking input space with memory space. The DDE operator transforms each $\boldsymbol{x}_{\langle i,j \rangle}$, refining feature representation to converge towards the global attractor. This process, governed by learnable time increments, captures both local variations and global patterns. As a result, the model transcends local information and approximates solutions in high-dimensional space. Specific details of the COS module are provided in **Algorithm** 1.

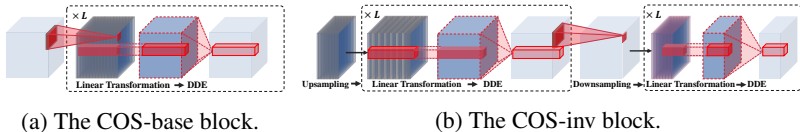

(a) The COS-base block.          (b) The COS-inv block.

Figure 3: The illustrations of the COS-base block (a) and the COS-inv block (b). $L$ represents the pre-defined discrete layers for the implementation of nmODEs.

**The basic and inverse residual blocks.** As shown in Fig. 3 (a), we retain the depthwise convolution and replace the pointwise convolution with our COS for inter-channel computation using DDE. Inspired by the inverted bottleneck block in MobileNetV2 [Sandler et al., 2018], we further design the COS-inv block in Fig. 3 (b). The main difference is that COS-inv places the COS module before the depthwise convolution for channel expansion, enhancing high-dimensional spatial computation and non-linear transformations by nmODEs. It then employs another COS to restore the channels, introducing an upsampling or downsampling process before the linear transformation. Specifically, bilinear interpolation is applied to each $\boldsymbol{x}_{\langle i,j \rangle}$, facilitating channel expansion or compression by considering surrounding pixels and maintaining image smoothness and detail.

| MobileNetV1 / Stride | Channels | Channelwise Param. | MobileODEV1 / Stride | Channels | Channelwise Param. | MobileODEV2 / Stride | Channels | Channelwise Param. |
|---|---|---|---|---|---|---|---|---|
| Conv2d / s2 | 3 | 3×32 | Conv2d / s2 | 3 | 3×32 | Conv2d / s2 | 3 | 3×32 |
| SeparableV1 / s1 | 32 | 32×64 | SeparableV1 / s1 | 32 | 32×64 | SeparableV1 / s1 | 32 | 32×64 |
| SeparableV1 / s2 | 64 | 64×128 | SeparableV1 / s2 | 64 | 64×144 | SeparableV1 / s2 | 64 | 64×144 |
| SeparableV1 / s1 | 128 | 128×128 | COS-base | 144 | 10×144+10 | COS-inv | 144 | 10×144×5+20 (e = 4) |
| SeparableV1 / s2 | 128 | 128×256 | SeparableV1 / s2 | 144 | 144×256 | SeparableV1 / s1 | 144 | 144×256 |
| SeparableV1 / s1 | 256 | 256×256 | COS-base | 256 | 10×256+10 | COS-inv | 256 | 10×144×5+20 (e = 4) |
| SeparableV1 / s2 | 256 | 256×512 | SeparableV1 / s2 | 256 | 256×529 | SeparableV1 / s2 | 256 | 256×529 |
| 5× SeparableV1 / s1 | 512 | 5×512×512 | 5× COS-base | 529 | 5×(10×529+10) | 5× COS-inv | 529 | 5×(10×144×5+20) (e = 4) |
| SeparableV1 / s2 | 512 | 512×1024 | SeparableV1 / s2 | 529 | 529×1024 | SeparableV1 / s2 | 529 | 529×1024 |
| SeparableV1 / s2 | 1024 | 1024×1024 | COS-base | 1024 | 10×1024+10 | COS-inv | 1024 | 10×1024×5+20 (e = 4) |
| Avg Pool | 1024 | - | Avg Pool | 1024 | - | Avg Pool | 1024 | - |
| FC | 1024 | 1024×200 | FC | 1024 | 1024×200 | FC | 1024 | 1024×200 |
| Softmax | 200 | - | Softmax | 200 | - | Softmax | 200 | - |
| **Total Param.** | - | **3.34M** | - | - | **0.97M**(↓ 71.0%) | - | - | **1.03M**(↓ 69.2%) |

Table 1: Comparison of architectures and parameters for MobileNetV1, MobileODEV1, and MobileODEV2 is presented. The red-highlighted numbers indicate the repeated use of the block.

**Detailed structures.** We have constructed MobileODEV1 and MobileODEV2, as shown in Tab. 1. SeparableV1 refers to the depthwise-separable convolution from MobileNetV1, where we retained certain pointwise convolutions for downsampling while maintaining linear computational capabilities. Both the COS-base and COS-inv blocks are followed by BatchNorm [Ioffe, 2015] and ReLU6 non-linearity. We made slight modifications to the input channels of the these blocks to enable square root calculations. An ablation study on discrete layer numbers is detailed in Sec. 4.4, leading us to set $L = 10$ for a balance between performance and speed. In MobileODEV1 and MobileODEV2, 8 depthwise-separable convolutions are replaced with the proposed COS-base and COS-inv blocks, respectively. The number of parameters for channelwise computation is shown in Tab. 1. COS in MobileODEV1 has only 0.04M parameters (↓ 98.36%) compared to the 2.44M of the replaced pointwise convolutions, and COS-inv has 0.10M (↓ 95.90%). Compared to MobileNetV1, MobileODEV1 reduces channelwise computational cost by 71.0%, while MobileODEV2 reduces it by 69.2%.

In COS-inv block, $e$ denotes the channel expansion factor. For pointwise convolution, increasing $e$ raises the computational cost by a factor of $e$ on top of the quadratic complexity (Tab. 2). In contrast, COS-inv has a linear increase in computational cost with respect to $e$. This feature enables COS-inv to scale to larger channel numbers without significantly increasing computational costs.

| Block | Formula | #Param.(eg. $C = 512, e = 4$) |
|---|---|---|
| Pointwise Conv | $C^2 \times e$ | $512^2 \times 4 = 1,048,576$ |
| COS-inv | $C \times e \times L + L$ | $512 \times 4 \times 10 + 10 = 20,490$ |

Table 2: Comparison of computational overhead between pointwise convolution and COS-inv with a channel expansion factor $e = 4$.

## 4 Experimental Results

In this section, we evaluate MobileODEV1 and MobileODEV2 incorporating self-attention mechanisms. We replace the final COS and COS-inv components with MobileViT blocks [Mehta and Rastegari, 2021], resulting in MobileODEV1+ViT and MobileODEV2+ViT. The MobileViT block integrates convolutional and transformer operations at low resolution, providing lightweight local and global perspective integration. our models achieving significant reductions in backbone size while improving performance. All simulations use fixed seeds for reproducibility, and models are implemented in PyTorch on a single NVIDIA 4090 GPU. For a fair comparison, we train all lightweight models from scratch without using pretrained parameters.

| Model | #Param. | CIFAR-10 | CIFAR-100 | IN-R | IN-tiny |
|---|---|---|---|---|---|
| **GhostNet** | 4.16M | 83.32 | 52.32 | 42.45 | 52.71 |
| **FastViT-T8** | 3.41M | 86.48 | 56.57 | 42.92 | 43.03 |
| **ShuffleNetV2** 1.0× | 1.46M | 92.31 | 71.54 | 46.72 | 59.97 |
| **ShuffleNetV2** 1.5× | 2.69M | 92.80 | 72.39 | 47.24 | 60.63 |
| **ShuffleNetV1** 1.0× | 1.10M | 92.29 | 70.55 | 45.49 | 59.21 |
| **ShuffleNetV1** 1.5× | 2.29M | 93.11 | 72.16 | 45.59 | 61.39 |
| **MobileNetV3-Large** | 4.41M | 93.55 | 73.72 | 44.72 | 60.93 |
| **MobileNetV4** | 2.75M | 92.43 | 74.14 | 44.39 | 61.10 |
| **MobileOne-s0** | 4.47M | 93.92 | 75.58 | 45.45 | 61.33 |
| **MobileNetV1** | 3.41M | 93.17 | 74.77 | 42.23 | 60.78 |
| **MobileODEV1** | 1.14M | 93.56 | 74.98 | 44.55 | 61.92 |
| **MobileNetV2** | 2.48M | 93.77 | 74.73 | 43.01 | 63.53 |
| **MobileODEV2** | 1.52M | 94.01 | 75.43 | 46.59 | 63.46 |
| **MobileViT** | 5.58M | 89.97 | 65.23 | 45.00 | 57.92 |
| **MobileODEV1+ViT** | 2.82M | 94.13 | 75.56 | 47.35 | 62.98 |
| **MobileViTV2** | 18.45M | 89.33 | 65.78 | 46.72 | 60.82 |
| **MobileODEV2+ViT** | 3.65M | **94.26** | **75.89** | **48.08** | **63.56** |

Table 3: Comparison of classification performance across various methods and datasets, with values indicating top-1 accuracy (%). Bold entries highlight superior performance. The parameters of the presented models are evaluated on the IN-R dataset.

## 4.1 Image Classification

**Dataset and experiment settings.** We validate our proposed methods on four different datasets, namely CIFAR-10/CIFAR-100 [Krizhevsky et al., 2009], ImageNet-R (IN-R) [Hendrycks et al., 2021a] and Tiny Imagenet (IN-tiny) [Le and Yang, 2015]. For CIFAR-10/100, which are well-known for classification tasks, we use a resolution of $32 \times 32$. IN-tiny has 200 classes, and each class contains 500 training images, 50 validation images at a resolution of $64 \times 64$. Additionally, we assess the robustness of our methods on IN-R dataset, which consists of 30K images from 200 ImageNet classes, rendered in diverse textures and styles (e.g., paintings, embroidery, etc.). We divide IN-R into training and testing sets at a $4 : 1$ ratio, using a resolution of $256 \times 256$.

In our experiments, all networks are using identical training recipes to ensure fair comparisons. The batch size is set to 32 images for CIFAR-10/CIFAR-100, while for IN-tiny and IN-R, it is set to 16 images. For CIFAR-10/CIFAR-100, as suggested in Haase and Amthor [2020], we remove the first and second pooling operations of MobileNets to obtain a final feature map of size $4 \times 4$. Our experimental setup is consistent with Mehta and Rastegari [2021]. Basic data augmentation techniques, including random resized cropping and horizontal flipping, are applied during training.

For the comparative experiments, MobileODEV1 and MobileODEV2 are evaluated against their baselines, MobileNetV1 [Howard, 2017] and MobileNetV2 [Sandler et al., 2018]. Additionally, they are compared with other classic lightweight networks that have been improved based on MobileNet, including GhostNet [Han et al., 2020], ShuffleNetV1 [Zhang et al., 2018], ShuffleNetV2 [Ma et al., 2018], MobileNetV3 [Howard et al., 2019], MobileNetV4 [Qin et al., 2025], and MobileOne [Vasu et al., 2023b]. Furthermore, with the integration of MobileViT blocks [Mehta and Rastegari, 2021], MobileNetV1+ViT and MobileNetV2+ViT are compared with MobileViT [Mehta and Rastegari, 2021], MobileViTV2 [Mehta and Rastegari, 2021], and FastViT-T8 [Vasu et al., 2023a].

**Results.** As shown in Tab. 3, MobileODEV1 and MobileODEV2 show significant accuracy improvements while keeping parameter counts low. MobileODEV1 has 1.14M parameters, which is 66.57% fewer than MobileNetV1 (3.41M), and MobileODEV2 has 1.52M parameters, 38.7% fewer than MobileNetV2 (2.48M). On CIFAR-10, MobileODEV1 outperforms MobileNetV1 by 0.39%, and MobileODEV2 surpasses MobileNetV2 by 0.24%. For CIFAR-100, MobileODEV1 improves over MobileNetV1 by 0.21%, while MobileODEV2 achieves a 0.7% gain over MobileNetV2. On IN-tiny and IN-R datasets, MobileODEV2+ViT exceeds the best results by 0.03% compared to MobileNetV2 and by 0.84% compared to ShuffleNetV2 $1.5\times$. Overall, MobileODEV2 consistently outperforms MobileODEV1 across all datasets, particularly in complex IN-R scenarios. This confirms the practical applicability of our methods and the superior learning capability of COS-inv over COS. Integrating MobileViT blocks [Mehta and Rastegari, 2021] enhances performance across all datasets. MobileODEV2+ViT consistently achieves the best results, highlighting the effectiveness of combining MobileODE with self-attention while minimizing computational overhead.

| Model | #Param. | PASCAL VOC2012 | ADE20K |
|---|---|---|---|
| MobileNetV3-Large | 6.69M | 52.11 | 26.82 |
| MobileNetV4 | 17.08M | 50.85 | 26.64 |
| MobileOne-s0 | 24.55M | 53.27 | 27.21 |
| MobileNetV1 | 20.33M | 49.39 | 25.96 |
| MobileODEV1 | **6.40M** | 51.73 | 26.88 |
| MobileNetV2 | 8.09M | 50.10 | 24.69 |
| MobileODEV2 | **6.39M** | 51.82 | 27.12 |
| MobileViT | 8.69M | 50.86 | 24.73 |
| MobileODEV1+ViT | 7.04M | 52.13 | 26.98 |
| MobileViTV2 | 35.38M | 47.31 | 24.25 |
| MobileODEV2+ViT | 7.48M | **53.35** | **27.32** |

Table 4: Comparison of semantic segmentation performance of DeepLabV3 with different backbones across various datasets, evaluated using mIOU (%). Model parameters are assessed on the ADE20K dataset.

| Model | #Param. | BUSI | | | FFE | | |
|---|---|---|---|---|---|---|---|
| | | mAP | $AP_{50}$ | $AP_{75}$ | mAP | $AP_{50}$ | $AP_{75}$ |
| MobileNetV3-Large | 3.07M | 35.61 | 67.80 | 30.84 | 62.82 | 95.44 | 73.50 |
| MobileNetV4 | 1.99M | 35.80 | **75.81** | 33.70 | 61.98 | 95.21 | 70.51 |
| MobileOne-s0 | 5.09M | 36.02 | 73.10 | 32.22 | 59.97 | 94.42 | 67.72 |
| MobileNetV1 | 3.90M | 30.71 | 61.13 | 30.02 | 61.85 | 94.02 | 71.82 |
| MobileODEV1 | **1.99M** | 32.50 | 64.73 | 28.94 | 61.88 | 91.35 | 73.13 |
| MobileNetV2 | 2.19M | 31.72 | 58.51 | 33.62 | 62.84 | 95.82 | 72.11 |
| MobileODEV2 | **1.41M** | 34.42 | 64.56 | 36.93 | 63.15 | 94.33 | 74.76 |
| MobileViT | 5.14M | 22.91 | 49.02 | 19.85 | 60.93 | 94.43 | 69.90 |
| MobileODEV1+ViT | 3.14M | 34.82 | 67.36 | 30.33 | 63.91 | 95.66 | 73.54 |
| MobileViTV2 | 18.55M | 35.91 | 62.64 | 37.72 | 66.10 | 94.41 | **78.93** |
| MobileODEV2+ViT | 6.15M | **37.71** | 66.40 | **37.82** | **66.52** | **95.74** | 77.52 |

Table 5: Comparison of object detection performance of SSDLite with different backbones on BUSI and FFE datasets. The parameters of the presented models are evaluated on the FFE dataset.

## 4.2 Semantic Segmentation

**Dataset and experiment settings.** We integrate MobileODE with DeepLabV3 [Chen, 2017], training MobileODE on PASCAL VOC 2012 [Everingham et al., 2015] dataset and ADE20K dataset [Zhou et al., 2019] from scratch, in accordance with standard training practices [Mehta and Rastegari, 2021]. All networks are trained for 200 epochs with a batch size of 16 images using a standard sampler. For the PASCAL VOC 2012 dataset, additional annotations from [Hariharan et al., 2011] are utilized. To substantially reduce the complexity of the segmentation head, we adjust the final output channels of MobileODEV1 and MobileODEV2 to 576 and 256, respectively, and eliminate the classification layer. This modification facilitates a more accurate evaluation of MobileODE's performance as a lightweight backbone network.

**Results.** As shown in Tab. 4, MobileODEs significantly improve performance with a much lower parameter count than other MobileNet variants. On PASCAL VOC 2012, MobileODEV1 improves over MobileNetV1 by 2.34%. MobileODEV2 surpasses MobileNetV2, achieving a 1.72% higher accuracy. When integrated with the MobileViT block [Mehta and Rastegari, 2021], MobileODEV1+ViT improves MobileViT by 1.27%, while MobileODEV2+ViT exceeds MobileViTV2 by 6.04%. On the ADE20K dataset, MobileODEV1 improves upon MobileNetV1 by 0.92%, using just 6.40M parameters versus MobileNetV1's 20.33M. MobileODEV2 surpasses MobileNetV2 by 2.43%. MobileODEV2+ViT achieves the highest accuracy just below MobileOne, recording 0.08% higher on PASCAL VOC 2012 and 0.11% higher on ADE20K, all while having 69.53% fewer parameters.

## 4.3 Object Detection

**Dataset and experiment settings.** We integrate MobileODE with the single-shot object detection (SSD) backbone [Liu et al., 2016] to develop an efficient object detection framework. We replace standard convolutions in the SSD head with separable convolutions, resulting in a lightweight model referred to as SSDLite [Mehta and Rastegari, 2021]. We train and validate our model's detection performance on the publicly available BUSI dataset [Al-Dhabyani et al., 2020] and Facial Feature Extraction (FFE) dataset [kag, 2025]. The BUSI dataset contains 437 images with benign lesions, 210 images with malignant lesions, and 133 normal images without lesions. The FFE dataset is a labeled dataset designed for the detection of various facial features, including eyebrows, eyes, nose, lips, and mustache-beard regions, with 457 images allocated for training and 126 images designated for validation. Model performance is evaluated using the mAP metric across IoU thresholds ranging from 0.50 to 0.95 (mAP@IoU $0.50 : 0.05 : 0.95$). Additionally, we report performance at specific IoU thresholds, including $AP_{50}$ and $AP_{75}$, to provide a more detailed evaluation.

**Results.** As shown in Tab. 5, on the BUSI dataset, MobileODEV1 outperforms MobileNetV1, achieving a 1.79% increase in AP and a 3.6% increase in $AP_{50}$. MobileODEV2 surpasses MobileNetV2 across all evaluation metrics, particularly in $AP_{50}$ ($\uparrow$ 6.05%). The performance of MobileODEV1+ViT and MobileODEV2+ViT significantly exceeds that of the original MobileViT model. For instance, MobileODEV1+ViT improves by 11.91% in AP and 18.3% in $AP_{50}$, while MobileODEV2+ViT surpasses MobileViTV2 by 1.8% in AP. For the FFE dataset, MobileODEV2+ViT outperforms MobileViTV2 with fewer parameters (6.15M vs. 18.55M). Overall, MobileODEV1 and MobileODEV2 perform exceptionally well as lightweight backbones for detection tasks, with the introduction of self-attention mechanisms providing a further boost in detection performance.

## 4.4 Ablation Study

**Impact of $L$.** We apply both learnable time increments $\Delta t$ and fixed time increments $\Delta t$ ($\Delta t = \frac{1}{L}$) to MobileODEV1 and MobileODEV2. As shown in Fig. 4 (a) and (b), it is evident that as the number of discrete layers increases, a fixed $\Delta t$ leads to unstable accuracy improvements, and even a decline in performance. This aligns with our expectation in a non-linear environment. In contrast, for the learnable $\Delta t$, the model accuracy exhibits an increasing trend. However, when $L$ is set to 10, the rate of improvement slowed. Additionally, by leveraging COS-inv blocks, MobileODEV2 demonstrates a more pronounced improvement in accuracy, following a similar trend.

**Impact of learnable $\Delta t$.** We explore the learning process of the learnable $\Delta t$ settings in MobileODEV2. In Fig. 4 (c), we present scatter plots of accuracy across all 200 categories of IN-R for each discrete layer in the final COS-inv module, with the initial setting of $L = 30$. The black points indicate the accuracy of the first discrete layer's output for each class, while the red points represent

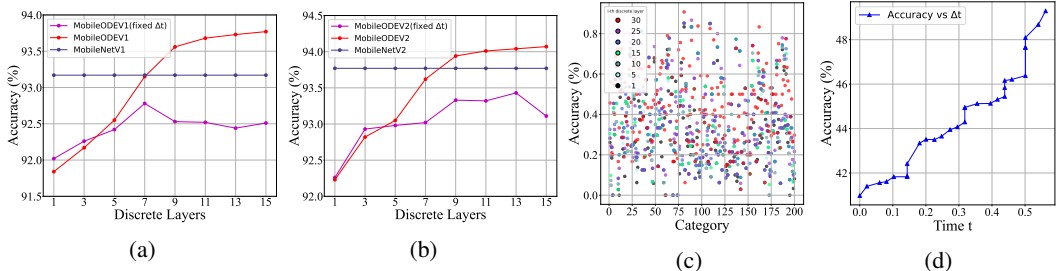

Figure 4: Study of the $L$ setting in MobileODEV1(a) and MobileODEV2(b), with learnable $\Delta t$ versus fixed $\Delta t$ on the CIFAR-10 dataset. (c) Accuracy across each category at each discrete layer. (d) Accuracy with respect to learnable time increments for 30-layer DDE.

the accuracy of the last discrete layer's output. A clear upward trend in accuracy is observed for each category with the increase in discrete layers, indicating that the COS design exhibits fine-grained perception. In Fig. 4 (d), we plot the values of the learnable $\Delta t$, with the spacing between the triangle markers proportional to the time increment. It is evident that the trend of accuracy improvement becomes more pronounced, highlighting the impact of the time increments accumulating across multiple layers effectively.

| e | #Param. | ACC |
|---|---|---|
| 1 | 0.61M | 93.57 |
| 4 | 1.27M | 94.01 |
| 9 | 1.77M | 91.04 |
| 16 | 2.46M | 89.57 |
| 25 | 3.35M | 88.02 |

Table 6: Study of $e$ settings.

| Model | FLOPs(M) | Latency(ms) |
|---|---|---|
| MobileNetV1 | 4.06 | 43.70 |
| MobileODEV1 | 7.09 | **24.54** |
| MobileODEV1 w/Runge-Kutta | 18.49 | 50.12 |
| MobileViT | 7.10 | 46.22 |
| MobileODEV1+ViT | 7.32 | **30.64** |
| MobileODEV1+ViT w/Runge-Kutta | 18.27 | 43.16 |

Table 7: Comparison of MobileODEV1 and MobileODEV1+ViT in terms of latency and FLOPs.

**Impact of $e$.** Tab. 6 presents the number of parameters and Top-1 accuracy (%) for models under different expansion factor $e$ settings, all evaluated on the CIFAR-10 dataset. Overall, the analysis reveals notable fluctuations in model performance as $e$ increases. Specifically, model performance significantly declines, highlighting the limitations of using bilinear interpolation for parameter-free channel expansion and contraction concerning the value of $e$.

**Computing efficiency.** In Tab. 7, MobileODEV1 exhibits a significantly lower latency of $24.54$ ms (batch size = 16) compared to the baseline models, despite a moderate increase in FLOPs to $7.09$ million. This reduction in latency indicates enhanced efficiency in processing. Although the increase in FLOPs is a common characteristic of ODE methods, such as those using the Runge-Kutta approach, our proposed methods have effectively addressed this challenge. Moreover, MobileODEV1+ViT maintains a competitive FLOP count of $7.32$ million while achieving a latency of $30.64$ ms, further demonstrating the effectiveness of our approach.

## 5  Conclusion

In this paper, we introduce MobileODE, an ultra-lightweight network based on the discrete nmODEs method, designed for various tasks. Our findings indicate a clear trend of improved performance as the number of discrete layers increases, emphasizing the potential for achieving state-of-the-art results. However, COS's sequential computation limits parallel processing, making indiscriminate increases in $L$ detrimental to inference speed. Therefore, a more balanced approach between performance and speed is essential. In future work, we plan to integrate Neural Architecture Search (NAS) from MobileNetV3-V4 into MobileODE to create a more efficient structure and further explore $L$ settings. We hope this research inspires advancements in lightweight network models.

## Acknowledgements

This work is supported in part by the National Natural Science Foundation of China (Grant No. 62495064 and 62206189).

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

# 6 Appendix

## 6.1 Training details for classification task

The loss function used cross-entropy with label smoothing (smoothing = 0.1), and optimization was performed with AdamW [Loshchilov, 2017], and a weight decay of 0.01. The learning rate started at 0.0002, increased to 0.002 for the first 2k iterations, and then decayed to 0.0002 using a cosine schedule [Loshchilov and Hutter, 2016]. The model was trained for a maximum of 200 epochs.

## 6.2 Training details for SSDLite and deeplabv3

Our experimental setup is consistent with Mehta and Rastegari [2021]. The hyperparameter settings for semantic segmentation task on PASCAL VOC 2012 dataset and object detection task are consistent. The AdamW optimizer with a weight decay of 0.01 is employed. The learning rate is first increased from 0.00009 to 0.0009 over the first 500 iterations, then annealed to $1e - 6$ using a cosine scheduler. For the ADE20K dataset, training uses the SGD optimizer with a momentum of 0.9 and a weight decay of 0.0001. A cosine annealing schedule is applied, reducing the learning rate from 0.02 to 0.0001.

## 6.3 How DDE adaptively modify each $\Delta t^l$.

Let the objective function be $L(\theta)$, where $\theta$ represents the model parameters. The learnable step size $\Delta t^l$ in DDE is optimized through backpropagation, and its gradient can be expressed as:

$$\frac{\partial L}{\partial \Delta t^l} = \frac{\partial L}{\partial y^{l+1}} \cdot \frac{\partial y^{l+1}}{\partial \Delta t^l} \tag{6}$$

where $y^{l+1} = y^l + \Delta t^l \cdot \dot{y}^l$. Expanding this gives:

$$\frac{\partial L}{\partial \Delta t_l} = \frac{\partial L}{\partial y^{l+1}} \cdot \dot{y}^l \tag{7}$$

Steep Regions: When $\|\dot{y}^l\|$ is large (high curvature), the absolute value of the gradient $\frac{\partial L}{\partial \Delta t^l}$ is large, forcing $\Delta t^l$ to decrease to reduce loss. Flat Regions: When $\|\dot{y}^l\|$ is small (low curvature), the absolute value of the gradient $\frac{\partial L}{\partial \Delta t^l}$ is small, allowing $\Delta t^l$ to increase to accelerate convergence.

## 6.4 Accuracy vs parameters on classification task

Tab. 8 complements Tab. 3 by providing additional details on parameter counts for CIFAR-10 and CIFAR-100, along with two sets of comparative experiments conducted using ResNet-110 and ResNet-353. Moreover, we add another challenging dataset ImageNet-A (IN-A) [Hendrycks et al., 2021b], which contains 7.5K images across 200, featuring natural adversarial examples designed to challenge existing models.

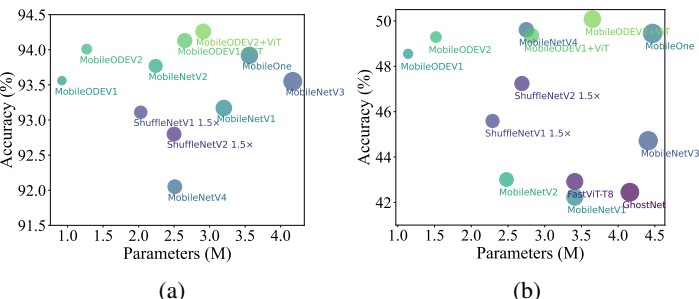

Figure 5: Visualization of accuracy results on the CIFAR-10 (a) and IN-R (b) datasets. The x-axis represents the number of parameters (lower is better), while the y-axis indicates accuracy (higher is better).

Fig. 5 visualizes the accuracy results on the CIFAR-10 and IN-R datasets. Our proposed models, MobileODEV1 and MobileODEV2, achieve impressive performance with a minimal number of parameters. Moreover, integrating self-attention mechanisms further enhances the results, leading to the best overall performance.

## 6.5 Licenses and copyrights across assets

1. **CIFAR-10/100**
   - Citation: Krizhevsky et al. [2009]
   - Asset Link: [https://www.cs.toronto.edu/~kriz/cifar.html]

2. **Tiny ImageNet (IN-tiny)**
   - Citation: Le and Yang [2015]
   - Asset Link: [https://huggingface.co/datasets/zh-plus/tiny-imagenet]

3. **ImageNet-R (IN-R)**
   - Citation: Hendrycks et al. [2021a]
   - Asset Link: [https://github.com/hendrycks/imagenet-r]
   - License:[https://github.com/hendrycks/imagenet-r?tab=MIT-1-ov-file]

4. **PASCAL VOC2012**
   - Citation: Everingham et al. [2015]
   - Asset Link: [http://host.robots.ox.ac.uk/pascal/VOC/voc2012/]

5. **ADE20K**
   - Citation: Zhou et al. [2019]
   - Asset Link: [https://ade20k.csail.mit.edu/index.html#Download]
   - License: [https://opensource.org/license/BSD-3-Clause]

6. **BUSI**
   - Citation: Al-Dhabyani et al. [2020]
   - Asset Link:[https://www.kaggle.com/datasets/sabahesaraki/breast-ultrasound-images-dataset]

7. **FFE**
   - Citation: kag [2025]
   - Asset Link:[https://www.kaggle.com/datasets/osmankagankurnaz/facial-feature-extraction-dataset]
   - License: [https://www.mit.edu/~amini/LICENSE.md]

## 6.6 Demo in edge computing platform

We have developed a breast screening demo to effectively apply the proposed lightweight MobileODE models. This framework consists of three stages. In the first stage, a binary classification distinguishes nodules from non-nodules (e.g., vessels, fat, muscle, etc.). Data identified as nodules are then passed to the second stage for detection. Images containing detected nodules, along with mask images generated from the detection boxes, are subsequently processed in the third stage to perform BIRADS grading at the nodule level (as opposed to the image level). This framework has been successfully deployed on an edge computing platform, as illustrated in Fig. 6. Since MobileODE is part of the components for BIRADS classification, we did not elaborate on this demo in the main text. To date, we have conducted screening trials across 11 hospitals (names withheld for anonymity policies), totaling 12,643 cases. With the assistance of six ultrasound physicians for secondary verification, we calculated a consistency accuracy of 94.4%.

For hardware,We use the Lenovo Legion R9000P laptop, equipped with an NVIDIA GeForce RTX 4060 GPU (8 GB VRAM), AMD Ryzen 9 7945HX CPU, 15.3 GiB DDR5 memory. It runs Ubuntu 20.04.4 LTS (Linux 5.15 kernel). During the running, GPU memory usage was 1708/8188 MiB, with a utilization of 19%, indicating steady but low resource usage.

| Model | #Param. | CIFAR-10 | #Param. | CIFAR-100 | IN-A |
|---|---|---|---|---|---|
| GhostNet | 3.91M | 83.32 | 4.03M | 52.32 | 7.02 |
| FastViT-T8 | 3.27M | 86.48 | 3.33M | 56.57 | 9.13 |
| ShuffleNetV2 1.0× | 1.27M | 92.31 | 1.36M | 71.54 | 9.48 |
| ShuffleNetV2 1.5× | 2.49M | 92.80 | 2.59M | 72.39 | 9.56 |
| ShuffleNetV1 1.0× | 0.93M | 92.29 | 1.01M | 70.55 | 9.41 |
| ShuffleNetV1 1.5× | 2.03M | 93.11 | 2.06M | 72.16 | 8.84 |
| MobileNetV3-Large | 4.17M | 93.55 | 4.29M | 73.72 | 10.07 |
| MobileNetV4 | 2.51M | 92.43 | 2.62M | 74.14 | 8.33 |
| MobileOne-s0 | 3.56M | 93.92 | 3.67M | 75.58 | 9.86 |
| ResNet-110 | 1.15M | 93.78 | 1.17M | 73.83 | - |
| ResNet-353 | 3.65M | 93.85 | 3.67M | 72.11 | - |
| MobileNetV1 | 3.20M | 93.17 | 3.29M | 74.77 | 8.42 |
| MobileODEV1 | **0.94M** | 93.56 | **1.00M** | 74.98 | 10.12 |
| MobileNetV2 | 2.24M | 93.77 | 2.35M | 74.73 | 8.90 |
| MobileODEV2 | **1.27M** | 94.01 | **1.36M** | 75.43 | 10.72 |
| MobileViT | 4.94M | 89.97 | 5.00M | 65.23 | 9.46 |
| MobileODEV1 +ViT | 2.65M | 94.13 | 2.75M | 75.56 | 10.94 |
| MobileViTV2 | 17.44M | 89.33 | 17.53M | 65.78 | 9.11 |
| MobileODEV2 +ViT | 3.28M | **94.26** | 3.40M | **75.89** | **10.95** |

Table 8: As a complement to Tab. 3, we provide the parameter counts for various models on the CIFAR-10 and CIFAR-100 datasets, respectively. We also present a new challenging benchmark IN-A to prove the generalizability of our method

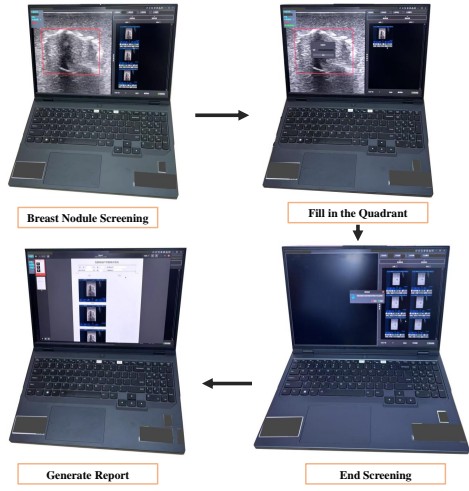

Figure 6: The applications in edge computing platform.

## 6.7 The difference between SeparableV1 and SeparableV2

We incorporated the depthwise-separable convolution and inverted residual structure proposed in MobileNetV1 and MobileNetV2, naming them SeparableV1 and SeparableV2, respectively. SeparableV1 consists of two separate layers: a lightweight depthwise convolution for spatial filtering and heavier $1 \times 1$ pointwise convolutions for feature generation. MobileNetV2 [Sandler et al., 2018] introduced the linear bottleneck and inverted residual structure, creating more efficient layer designs by leveraging the low-rank nature of the problem. SeparableV2 is defined by a $1 \times 1$ expansion convolution, followed by depthwise convolutions and a $1 \times 1$ projection layer. A residual connection is applied only when the input and output have the same number of channels. This structure maintains a compact representation at both the input and output while expanding internally to a higher-dimensional feature space, thereby enhancing the expressiveness of nonlinear per-channel transformations.

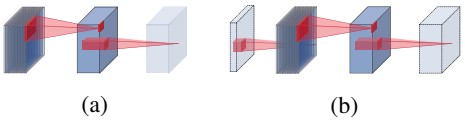

(a)        (b)

Figure 7: (a) SeparableV1: MobileNetV1 layer (depthwise-separable convolution). (b) SeparableV2: MobileNetV2 layer (Inverted Residual and Linear Bottleneck). Each block consists of a narrow input and output (bottleneck) without nonlinearity, followed by an expansion to a much higher-dimensional space and a projection to the output. The residual connection links the bottleneck rather than the expanded representation.

## 6.8 Broader Impacts

This work advances the development of lightweight convolutional neural networks (CNNs) through the discretization Ordinary Differential Equations (ODEs), showcasing significant potential for positive societal impact. The proposed MobileODEs enable the execution of complex visual tasks on resource-limited devices, such as smartphones and edge computing platforms (described in Sec. 6.6). Despite the improved accuracy of the model in various tasks, lightweight networks may generate inaccurate results in some cases, which, if not rigorously validated, could lead to the spread of misinformation. In critical areas like healthcare and law, such misinformation could have serious consequences. Therefore, while the technology shows great promise, responsible deployment and further research into alignment and safety remain crucial.

## 6.9 Qualitative results on the task of object detection

Fig.8 and Fig.9 showcase the performance of SSDLite equipped with MobileODEV1 and Mo-bileODEV2, demonstrating successful detection of a diverse range of objects, including facial features and breast tumors. These results underscore the strong generalization capabilities of our proposed models across various mobile network variants. Furthermore, the integration of MobileODE enhances the model's ability to adapt to complex tasks, making it suitable for real-world applications that demand both accuracy and efficiency.

## 6.10 Qualitative results on the task of semantic segmentation

Fig.10 and Fig.11 present the visualizations of semantic segmentation results obtained using DeepLabv3-MobileODEV2+ViT on the PASCAL VOC 2012 and ADE20K datasets, respectively. The left column displays the input RGB images, the middle column shows the predicted segmentation masks, and the right column overlays the segmentation masks onto the RGB images. The qualitative results demonstrate a strong understanding of scene semantics, highlighting the effectiveness and robustness of our proposed method across different datasets and complex scenarios.

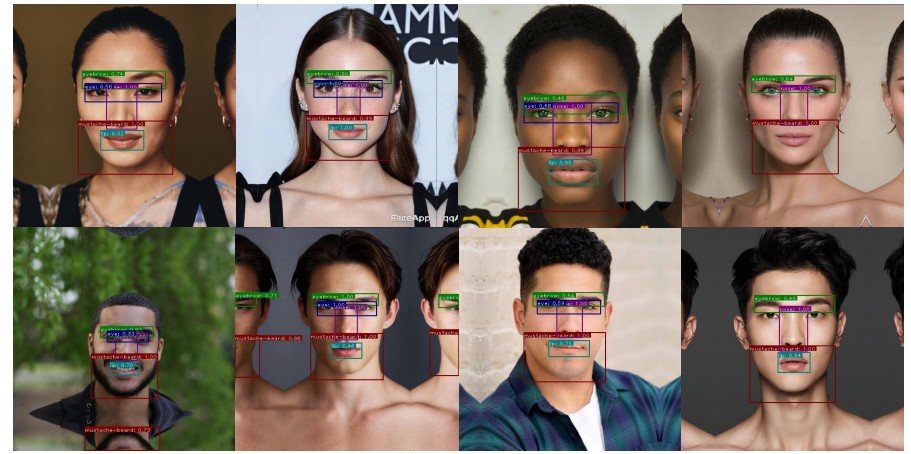

Figure 8: The object detection results of SSDLite-MobileODEV1 (first row) and SSDLite-MobileODEV2 (second row) on the Facial Feature Extraction Dataset validation set.

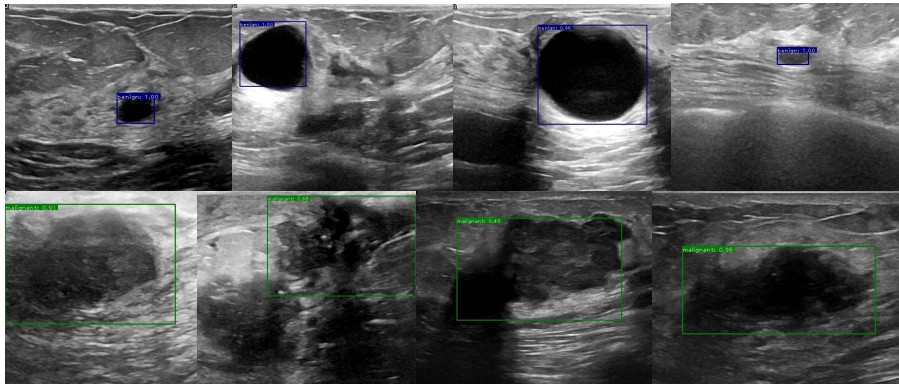

Figure 9: The object detection results of SSDLite-MobileODEV1 on benign tumors (first row) and malignant tumors (second row) demonstrate the performance on the BUSI dataset.

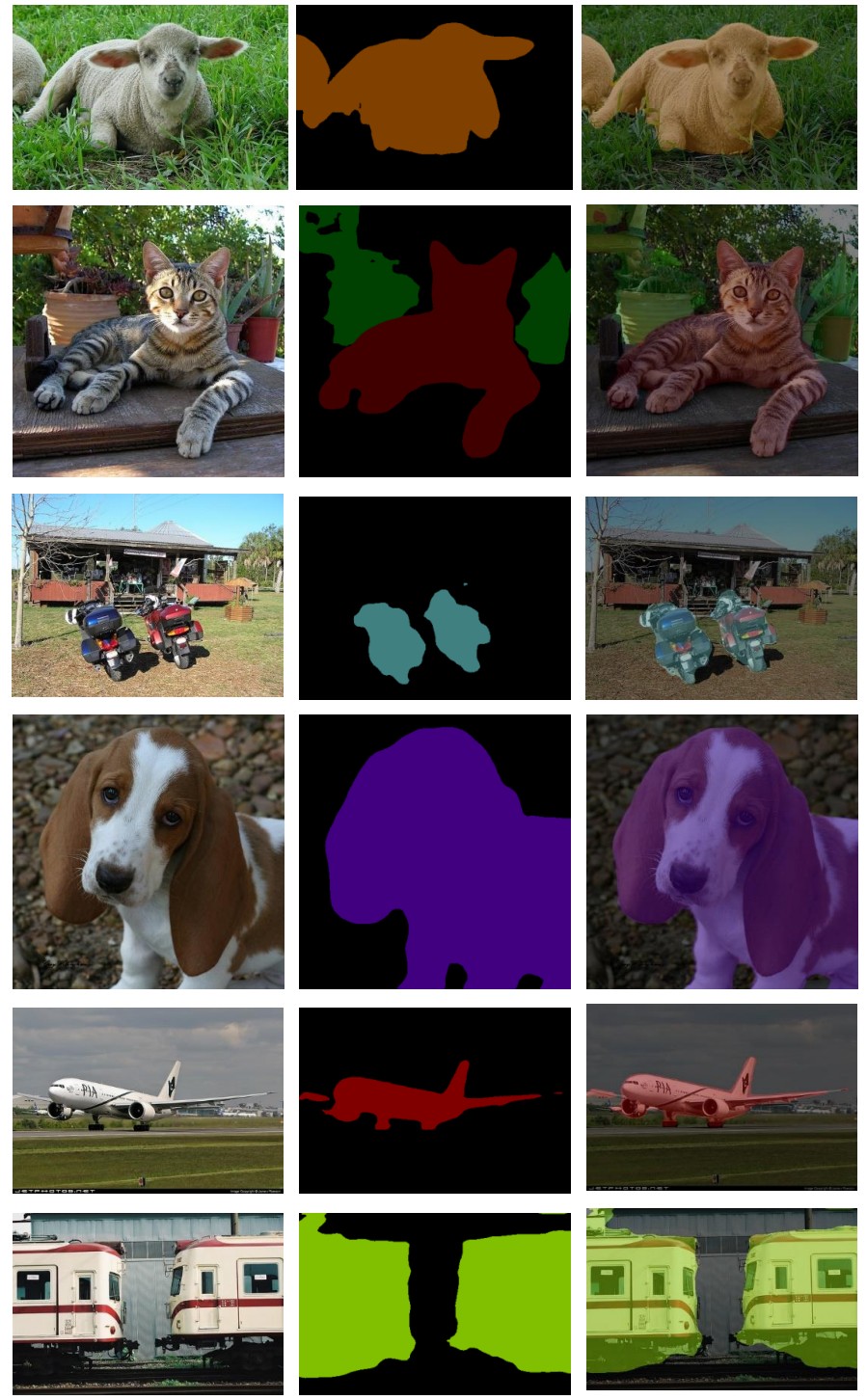

Figure 10: The semantic segmentation results of SSDLite-MobileODEV2+ViT on PASCAL VOC 2012 dataset.

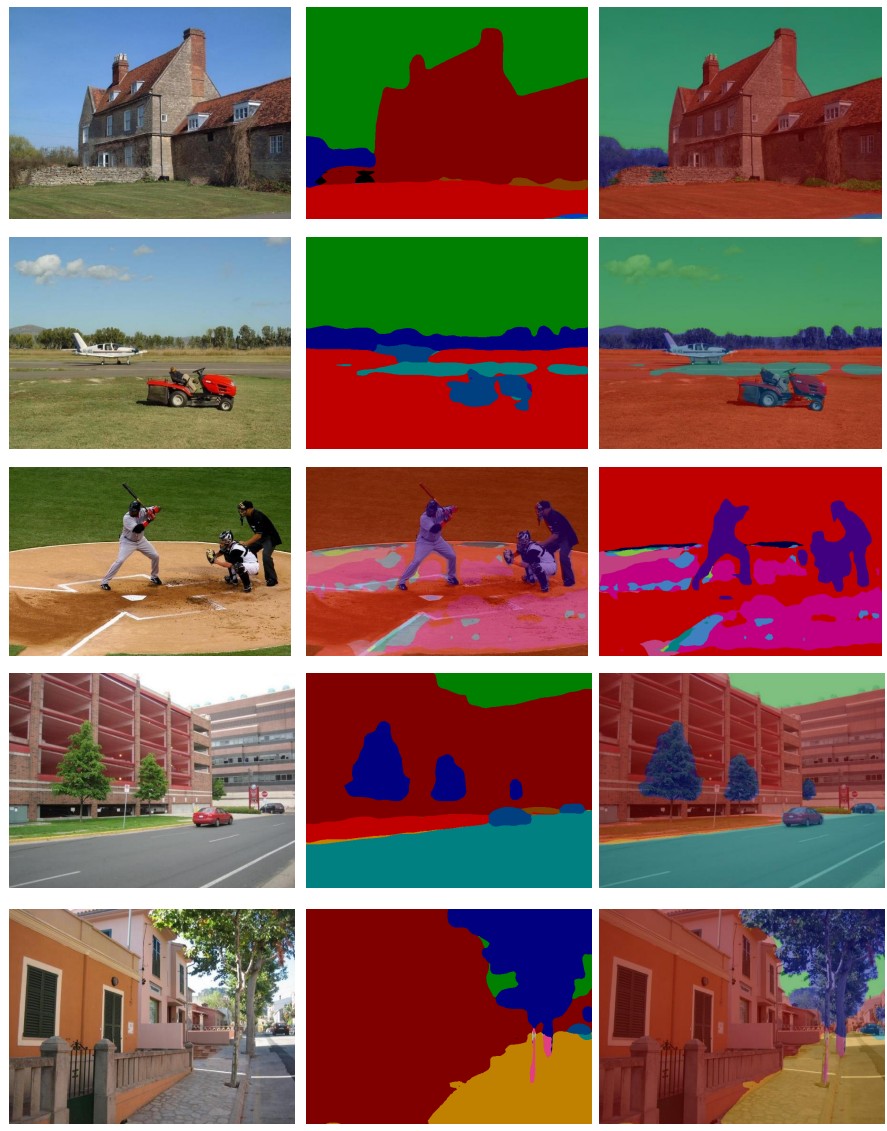

Figure 11: The semantic segmentation results of SSDLite-MobileODEV2+ViT on ADE20K dataset.

