# OpenReview forum: "MobileODE: An Extra Lightweight Network"
_NeurIPS.cc/2025/Conference — NeurIPS 2025 poster_

### Official Review · Reviewer_Tqqz · 2025-06-24

**Clarity:** 3
**Significance:** 3
**Originality:** 3
**Rating:** 4
**Confidence:** 4

**Summary:**

The paper introduces MobileODE, a novel lightweight CNN architecture that replaces pointwise convolution in depthwise-separable convolution with a Channelwise ODE Solver (COS) module based on discrete neural memory ODEs (nmODEs). By modeling channel interactions via a learnable time-increment Euler discretization (DDE), MobileODE achieves 98.36% parameter reduction compared to conventional pointwise convolution. Two variants, MobileODEV1 (0.97M parameters) and MobileODEV2 (1.03M), outperform MobileNetV1/V2 and other lightweight networks (e.g., GhostNet, ShuffleNet) on image classification, object detection, and segmentation tasks while maintaining low computational overhead.

**Questions:**

- What the performance difference between DDE and higher-order solvers like Runge-Kutta? Table 7 only shows FLOPs and latency.
- It is possible to apply COS modules to NLP or speech?
- Why omit MobileNetV3/V4, MobileOne? How does COS compare to dynamic convolution or attention-based channel mixers (e.g., in EfficientFormerV2)?
- COS reduces parameters but increases FLOPs (Tab 7). Is this favorable on edge devices? Are there any energy consumption support on edge devices?
- Why $\sqrt{C}$ factorization for $\Phi$? Ablate against linear (C×k) or low-rank (C×r×r) alternatives.
- Why not test on full ImageNet-1K? Accuracy on IN-tiny (64×64) may not reflect real-world performance.

I give a borderline score currently, may change my rating once the authors answered my concerns.

**Ethical Concerns:**

["NO or VERY MINOR ethics concerns only"]

**Final Justification:**

Increasing score from 3=>4 based on rebuttal, I think this paper is acceptable for the main conference.

**Limitations:**

Yes

**Quality:**

3

**Strengths And Weaknesses:**

Strengths
+ Novel architecture: the Discrete Derivative Estimator (DDE) with learnable time-increments yields more stable gradients and higher accuracy than fixed-step ODE solvers, while the Channel-wise ODE Solver (COS) replaces 1 × 1 point-wise convolutions, dramatically cutting channel-interaction parameters relative to MobileNet V1/V2.
+ Consistently strong empirical results across image classification, object detection, and segmentation.
+ The ODE-based variants match or surpass MobileNet latency despite using far fewer parameters.

Weaknesses
- Performance gains taper off at large expansion factors (e ≥ 9 in Table 6), hinting that the bilinear interpolation used for channel reshaping may bottleneck scalability.
- COS updates channels sequentially (Algorithm 1), which limits parallelism; for long ODE trajectories (large L) this can increase inference latency.
- The “extra-lightweight” claim is overstated: ShuffleNet V1 (1.1 M parameters) is still smaller than MobileODE-V1 (1.14 M) as shown in Table 3.

---

> ### Author Rebuttal · Authors · 2025-07-26
>
> We thank the reviewer for the valuable feedback and insightful questions. We appreciate the recognition of our contributions and the constructive suggestions, which have helped us further improve the quality and clarity of our work. Below, we address each concern in detail.
>
> ---
>
> ## Q1: What is the performance difference between DDE and higher-order solvers like Runge-Kutta?
>
> The DDE algorithm is currently based on discretizing nmODE with the Euler method. Technically, it also supports all higher-order solvers. We conducted further experiments on MobileODEV1 with other higher-order solvers on CIFAR-10, as shown in the table below:
>
> | Method                            | ACC    | FLOPs  | Latency (bs=16)  |
> |------------------------------------|:------:|:----------:|:------------:|
> | MobileODEV1 | 93.56% | 7.09 M  | 24.54 ms  |
> | MobileODEV1 w/ Runge-Kutta | 91.20% | 21.51 M   | 241.46 ms |
> | MobileODEV1 w/ Adams Bashforth | 90.66% | 14.89 M  | 158.29 ms  |
> | MobileODEV1 w/ Adams-Moulton | 91.17%  | 8.27 M | 104.32 ms |
> | MobileODEV1 w/ Itô–Taylor | 87.00%  |  11.58 M | 130.44 ms |
>
> While higher-order solvers like Runge-Kutta can theoretically improve accuracy, our preliminary experiments show that, in the context of lightweight discrete ODE models, their additional computational overhead does not translate to significant accuracy gains compared to DDE. It is very important for us to include these results in Table 7 in the revision.
>
> ---
>
> ## Q2: Is it possible to apply COS modules to NLP or speech?
>
> Excellent question! The COS module is a general mechanism designed for modeling interactions among feature channels. At its core, it functions similarly to an RNN block, where $y^l$ can be interpreted as the hidden state at time step $l$. This design makes COS inherently adaptable to non-vision domains such as NLP and speech.
>
> For instance, in Transformer-based models, COS could potentially be leveraged to capture token-wise or feature-wise dependencies, enhancing the model’s ability to represent complex relationships within sequences. We also encourage exploring the use of COS modules in various sequence modeling tasks beyond vision, as their flexible structure may offer benefits across a wide range of applications.
>
> Adapting COS to these domains would require redefining the dynamics over ODE or time steps, which we consider a promising direction for future work.
>
> ---
>
> ## Q3: Why omit MobileNetV3/V4, MobileOne?
>
> We define MobileNetV3/V4 and MobileOne as our baselines across three tasks. For the reason why ODEV variants are not provided, please refer to Q1 of Reviewer J9oG.
>
> ---
>
> ## Q4: How does COS compare to dynamic convolution or attention-based channel mixers (e.g., in EfficientFormerV2)?
>
> We compare COS to dynamic convolution and attention-based channel mixers in EfficientFormerV2 on CIFAR-10, as shown in the following table. DY-MobileNetV1 denotes replacing COS modules with dynamic convolutions, $K$ means convolution kernel number. SA-MobileNetV1 denotes replacing COS modules with the attention downsampling proposed in EfficientFormerV2 for mobile devices.
>
> | Method             | ACC    | Param. | FLOPs | Latency (bs=16) |
> |--------------------|:------:|:------:|:----------:|:------------:|
> | MobileODEV1        | 93.56% | 1.14 M   |7.09 M  | 24.54 ms |
> | DY-MobileNetV1 (K=2) | 90.63%  | 6.67 M    |  2.28M | 76.84 ms  |
> | DY-MobileNetV1 (K=4) | 90.29%  | 12.08 M    | 2.41M |  78.12 ms |
> | SA-MobileNetV1     | 93.52%  | 6.51 M    |  8.14G | 167.09 ms |
>
> ---
>
> ## Q5: COS reduces parameters but increases FLOPs (Tab 7). Is this favorable on edge devices? Are there any energy consumption measurements on edge devices?
>
> In Appendix 6.7, our practical deployment shows that this trade-off is favorable for edge devices. We have implemented our lightweight MobileODE-based breast screening system on a Lenovo Legion R9000P laptop (NVIDIA GeForce RTX 4060 GPU with 8GB VRAM, AMD Ryzen 9 7945HX CPU, 15.3GiB DDR5 RAM), running Ubuntu 20.04.4 LTS. During operation, GPU memory usage was 1708/8188 MiB and GPU utilization was only 19%, indicating low and stable resource consumption.
>
> Although we have not directly measured energy consumption, the low hardware usage suggests that the increased FLOPs do not adversely affect deployment efficiency on edge platforms.
>
> ---
>
> ## Q6: Why $\sqrt{C}$ factorization for $\Phi$? Ablate against linear (C×k) or low-rank (C×r×r) alternatives.
>
> Thank you for your question regarding the use of $\sqrt{C} × \sqrt{C}$ reshaping in our design. We would like to clarify that our approach does not involve matrix factorization or low-rank decomposition of a parameterized transformation matrix $\Phi$. Specifically, we reshape the 1D channel feature vector of size $C$ into a 2D matrix of size $\sqrt{C} × \sqrt{C}$ solely for the channel-view of applying subsequent certain operations (such as batch normalization and bilinear interpolation in MobileODEv2).
>
> Therefore, the comparison with linear (C×k) or low-rank (C×r×r) alternatives is not directly applicable. We will clarify this design choice and its motivation in the revised manuscript to avoid confusion.
>
> ---
>
> ## Q7: Why not test on full ImageNet-1K? Accuracy on IN-tiny (64×64) may not reflect real-world performance.
>
> Please refer to Q1 of Reviewer MTGp.
>
> ---
>
> # Concerns in the weakness.
>
> ## W1: The bilinear interpolation used for channel reshaping may bottleneck scalability.
>
> Thank you for highlighting this issue. We agree that as the expansion factor $e$ increases, bilinear interpolation may lead to information loss between channels and diminishing performance gains. The ablation study on $e$ in our work is intended to demonstrate that simply increasing $e$ for performance improvement should not be the main focus of MobileODEV2, which aligns with the reviewer's observation.
>
> ## W2: COS updates channels sequentially (Algorithm 1), which limits parallelism; for long ODE trajectories (large L) this can increase inference latency.
>
> The sequential channel update in COS is motivated by the explicit Euler discretization in ODE solvers, which is designed to capture dependencies across channels. While this sequential nature can limit full parallelism—potentially increasing inference latency for long ODE trajectories (large $L$)—our experiments show that the relationship between $L$ and latency is approximately linear, as demonstrated in Table Q2 for Reviewer MTGp. Thanks to the lightweight design of our architecture and the significant reduction in overall model parameters, the actual increase in inference time remains modest in our benchmarks. Moreover, we anticipate that future advances in hardware and software (e.g., pipelined or block-parallel COS updates) could further mitigate this limitation, offering even greater efficiency.
>
> ## W3: The "extra-lightweight" claim is overstated: ShuffleNet V1 (1.1 M parameters) is still smaller than MobileODE-V1 (1.14 M) as shown in Table 3.
>
> We acknowledge that, as shown in Table 3, ShuffleNet V1 (1.1M parameters) has a slightly lower parameter count than MobileODE-V1 (1.14M). However, our “extra-lightweight” claim is primarily based on the **flexibility** of the MobileODE architecture with respect to the $L$ setting. As demonstrated in the table below, MobileODE can be scaled down to just 0.885M parameters when $L=3$, while achieving performance closely matching that of ShuffleNet V1 (1.0×) on CIFAR-10.
>
> | Model                 | Discrete Layers    | Param.| Top-1 Acc (%) |
> |-----------------------|:------------:|:---------------:|:---------------:|
> | ShuffleNet V1 |\ | 0.93 M| 92.29|
> | MobileODEV1 | $L=3$ | 0.885 M |92.24 |
> | MobileODEV1| $L=5$ | 0.893 M| 92.52 |
> | MobileODEV1| $L=7$ |0.902 M| 93.16 |
> | MobileODEV1| $L=10$ |0.94 M |93.56 |
>
> Therefore, MobileODE offers not only competitive parameter efficiency but also **scalable flexibility**—allowing it to achieve even smaller sizes without significant performance loss. This adaptability makes MobileODE particularly suitable for resource-constrained scenarios and further supports our “extra-lightweight” claim.

---

> > ### Comment · Reviewer_Tqqz · 2025-08-05
> >
> > I have read the feedback and appreciate the detailed response by the authors.
> > The rebuttal answers most of my concerns. I would like to increase my rating accordingly.

---

### Official Review · Reviewer_J9oG · 2025-06-27

**Clarity:** 4
**Significance:** 3
**Originality:** 3
**Rating:** 5
**Confidence:** 4

**Summary:**

This paper proposes a method for making MobileNet CNNs more lightweight by using Ordinary Differential Equations as a replacement of the pointwise convolution in depthwise seperable convolutions. The authors apply their technique to MobileNetV1 and MobileNetV2 and provide experiments for classification tasks, semantic segmentation, and objects detection mainly using #Params besides Accuracy/mIOU/mAP to back up their claim.

**Questions:**

- Why did the authors provide ODEV variants for MobileNetV1 and V2, but not for the more recent versions, V3 and V4? Instead, they only used these for comparison in the experiments.

- Minor question regarding an inconsistency between the paper and the provided source code: In lines 144 ff. of the paper, the authors explicitely state that "These increments are then passed through a ReLU6 activation function to ensure they remain non-negative and to limit their output range." However, in the provided source code (cvnets/modules/mobilenetv2.py, line 259, CustomActivation class), something slightly different seems to be done (torch.maximum(torch.tensor(self.epsilon, device=x.device), x)), with the ReLU6 part being commented out above (line 258). While this implementation is basically a slightly modified version of ReLU and should still ensure that the values in x remain non-negative (given that the default value of epsilon is set to 1/num_layers), it does not enforce an upper limit on x, as ReLU6 does, but instead leaves the values in x unbounded.

**Ethical Concerns:**

["NO or VERY MINOR ethics concerns only"]

**Final Justification:**

The rebuttal I received from the authors answered my remaining questions sufficiently. Considering the rebuttals received by the other reviewers, which in my opinion answered their concerns detailed and transparantly, I will remain with my inital score of acceptance.

I want to add, that while I see the argument of reviewer mANM in that the performance degradation on TPUs might indicate that the proposed efficiency gains are not immediate and universal across all accelerators, I would personally argue that this is rarly the case for any algorithm since especially in high performance computing often extremly target specific adaptions are necessary to exploit different hardware architecture as best as possible, e.g., by explicitely using specific instruction set extensions which might not ne utilize automatically. So I would tend to agree with the response of the authors in that this insufficiency on TPUs could probably be improved by some target specific optimization and is not the result of a fundamental flaw in the proposed approach.

**Limitations:**

yes

**Paper Formatting Concerns:**

I found no major formatting issues

**Quality:**

4

**Strengths And Weaknesses:**

The substantial experiments indicate that the proposed COS module significantly reduces model parameters while even outperforming the AI performance achieved by the respective baselines (MobileNetV1 and V2), as well as more recent MobileNet variants (e.g. MobileNetV3, V4 and ShuffleNet). The high quality of the method, the explanation of its mathematical background, the hyperparameter ablation study of $L$, $\delta_t$ and $e$, and the provided source code make this a significant and novel contribution.

The part of the evaluation that focuses on computing efficiency is rather brief, especially since, as the authors also state, ODE methods have a high computational complexity. At the very least, the authors could specify the type of hardware on which the latency numbers were achieved and whether they represent averaged execution times. If so, they should also provide the standard deviation and clarify whether the measurements were taken per sample or per batch.

Figure 2: The visual difference between (a) and (b) is very subtle. This could be improved by introducing a colour to guide the reader to the different delta t values.

Minor inconsistency: in subfigure c, "Accuracy (%)" is defined between 0.0 and 1.0, but this should be between 0 and 100%, as in the rest of the paper.

---

> ### Author Rebuttal · Authors · 2025-07-25
>
> We thank the reviewer for these insightful questions, which will help guide our future research directions. We also appreciate the reviewer’s comments on our writing, which have improved the clarity and rigor of our paper.
>
> ---
>
> ## Q1: Why not provide ODEV variants for MobileNetV3 and V4?
>
> MobileNetV3 and V4 are fundamentally built upon the backbone architectures of MobileNetV1 and V2. Their improvements mainly involve incremental tricks and optimizations, rather than substantial structural changes to the core architecture. This is why our comparisons focus on V1 and V2, as they represent the foundational designs of the MobileNet series.
>
> Importantly, most of the tricks introduced in V3 and V4 can also be integrated into MobileODE. Exploring the incorporation of these enhancements, as well as further improving MobileODE with techniques such as NAS (Neural Architecture Search) and knowledge distillation, is an active area of our ongoing research.
>
> ---
>
> ## Q2: Inconsistency between the paper and the provided source code.
>
> We thank the reviewer for carefully pointing out the discrepancy between the manuscript and the provided code regarding the activation function applied after the increment operation. In fact, we have implemented two versions of the activation: one using ReLU6, as described in the paper, and another using `torch.maximum(epsilon, x)`. We apologize for not clearly documenting this in the README file.
>
> During our ongoing experiments with large step sizes $L$, we observed that enforcing non-negativity with `torch.maximum(epsilon, x)` helps prevent invalid step sizes and achieves comparable, and in some cases slightly better, numerical stability compared to ReLU6. We will update our findings in the new appendix of the manuscript and revise the code documentation to clarify this point and ensure consistency.
>
> ---
> # Concerns in the weakness.
>
> ## W1: Hardware specifications for latency measurements.
>
> All latency measurements were performed on a single NVIDIA RTX 4090 GPU. The reported latency represents the average inference time per batch, with a batch size of 16.
>
> Additionally, we explore the impact of the hardware platform, as detailed in Table Q1 of Reviewer mANM. The results show that our model consistently outperforms MobileNetV1 in latency on both CPU and GPU. Furthermore, performance can be further optimized on TPU, while also revealing additional optimization potential in MobileODEs.
>
> ---
>
> ## W2: Writing issues
>
> In the revised manuscript, we have updated Figure 2 by introducing distinct colors for different $\Delta t$ values to improve clarity.
>
> Additionally, we have corrected the axis in subfigure (c) of Figure 4 to range from 0 to 100% in the revised version.

---

> > ### Comment · Reviewer_J9oG · 2025-08-03
> >
> > Thank you for your rebuttal. I have no further questions about the paper at the moment.

---

### Official Review · Reviewer_mANM · 2025-07-03

**Clarity:** 3
**Significance:** 3
**Originality:** 3
**Rating:** 4
**Confidence:** 3

**Summary:**

This paper introduces MobileODE, a lightweight network architecture designed to improve the efficiency of depthwise-separable convolutions. The core idea is to replace the inefficient pointwise convolution component with a novel, parameter-efficient module named the Channelwise ODE Solver (COS). The COS is based on a simple yet efficient direct differentiation Euler algorithm. The paper introduces two variants of COS-based blocks (COS-base and COS-inv), and evaluates their performance on various tasks. The results show a significant reduction in parameters for channel interaction while achieving competitive or even improved accuracy.

**Questions:**

Please solve the questions in weaknesses.

**Ethical Concerns:**

["NO or VERY MINOR ethics concerns only"]

**Final Justification:**

The authors' rebuttal addressed my initial questions but also highlighted a key limitation. The new results show a performance degradation on TPUs, indicating the proposed efficiency gains are not universal across modern accelerators. This hardware dependency limits the overall impact and practical value of the contribution. Therefore, though the paper is technically sound, it has significant limitations in its general applicability, and I would keep my original score of "Borderline accept".

**Limitations:**

Yes.

**Paper Formatting Concerns:**

N/A.

**Quality:**

3

**Strengths And Weaknesses:**

**Strengths:**
- The paper addresses an important problem.
- The proposed method is evaluated comprehensively across different tasks and datasets, which helps demonstrate its generality.
- Experimental results are promising in terms of both model size reduction and accuracy.

**Weaknesses:**
- The experimental details for the latency benchmarks are insufficient. Since the proposed method increases FLOPs but is reported to be faster, it is important to clarify the evaluation setup (e.g., CPU/GPU model, memory) and understand if the reported latency improvements are specific to one type of hardware or if they generalize to other common accelerators like GPUs, NPUs, and TPUs, which are highly optimized for standard convolutions.
- The motivation could be made stronger. For example, providing a breakdown of the computational cost or latency contribution for depthwise and pointwise convolutions would better highlight the inefficiency of pointwise convolution and justify the design choice. Similarly, showing the latency before and after replacing pointwise convolution would help quantify the benefit of the proposed method more clearly.

---

> ### Author Rebuttal · Authors · 2025-07-28
>
> ## Q1: On Latency Benchmark Details.
>
> We appreciate the reviewer’s suggestion to provide a more detailed description of our latency benchmarks. As shown in the table, our model consistently outperforms MobileNetV1 in latency on both CPU and GPU. On the TPU, however, it is slightly slower, which we attribute to the sequential computation pattern of the COS module. We believe this overhead can be mitigated through hardware‑specific optimizations, especially as future accelerators become better optimized for directly differentiable models such as COS.
>
> To address the reviewer’s concern, we will include additional details on the hardware configurations and the benchmarking protocol in the revised manuscript.
>
> | Model       | Hardware Platform              | Latency (bs=16) | Memory Usage (%) |
> | ----------- | ------------------------------ | :------------------: | :---------------: |
> | MobileNetV1 | Intel(R) Xeon(R) CPU @ 2.20GHz |              166.67 ms |              9.2 |
> | MobileODEV1 | Intel(R) Xeon(R) CPU @ 2.20GHz |              148.61 ms |              5.2 |
> | MobileNetV1 | NVIDIA GeForce RTX 4090        |               33.70 ms |              3.3 |
> | MobileODEV1 | NVIDIA GeForce RTX 4090        |               24.54 ms |              4.9 |
> | MobileNetV1 | TPU v2-8                       |                3.28 ms |              0.6 |
> | MobileODEV1 | TPU v2-8                       |                8.37 ms |              0.7 |
>
> ## Q2: On Motivation and Ablation Studies.
>
> Thank you for your insightful feedback. In response, we further investigate the latency of depthwise convolution (dw) and pointwise convolution (pw) on a single RTX 4090 GPU to provide a more detailed comparison. The following table, which aligns with Table 1 in our paper, clearly demonstrates that the original pointwise convolution consumes a significant portion of the computational resources, while our COS module accelerates this process.
>
> Furthermore, we explore the impact of replacing pointwise convolution with COS-based blocks, particularly in the context of other channel computing modules and higher-order solvers. As detailed in Table Q1 and Q4 of Reviewer Tqqz, the results highlight how the COS module enhances both accuracy and computational efficiency.
>
> | Layer             | MobileODEV1 Latency (ms, bs=16) | MobileNetV1 Latency (ms, bs=16) |
> | ----------------- | :-------------------------------: | :------------------------: |
> | SeparableV1/s1/dw | 0.023                           | 0.030                    |
> | SeparableV1/s1/pw | 0.049                           | 0.054                    |
> | SeparableV1/s2/dw | 0.023                           | 0.029                    |
> | SeparableV1/s2/pw | 0.045                           | 0.052                    |
> | SeparableV1/s1/dw | 0.021                           | 0.027                    |
> | SeparableV1/s1/pw | 0.044                           | 0.053                    |
> | SeparableV1/s2/dw | 0.022                           | 0.025                    |
> | SeparableV1/s2/pw | 0.044                           | 0.053                    |
> | SeparableV1/s1/dw | 0.021                           | 0.028                    |
> | SeparableV1/s1/pw | 0.044                           | 0.053                    |
> | SeparableV1/s2/dw | 0.021                           | 0.024                    |
> | SeparableV1/s2/pw | 0.042                           | 0.054                    |
> | SeparableV1/s1/dw | 0.021                           | 0.028                    |
> | SeparableV1/s1/pw | 0.043                           | 0.054                    |
> | SeparableV1/s1/dw | 0.023                           | 0.024                    |
> | SeparableV1/s1/pw | 0.043                           | 0.052                    |
> | SeparableV1/s1/dw | 0.021                           | 0.025                    |
> | SeparableV1/s1/pw | 0.042                           | 0.053                    |
> | SeparableV1/s1/dw | 0.021                           | 0.023                    |
> | SeparableV1/s1/pw | 0.043                           | 0.052                    |
> | SeparableV1/s1/dw | 0.021                           | 0.023                    |
> | SeparableV1/s1/pw | 0.043                           | 0.052                    |
> | SeparableV1/s2/dw | 0.021                           | 0.023                    |
> | SeparableV1/s2/pw | 0.043                           | 0.054                    |
> | SeparableV1/s1/dw | 0.022                           | 0.027                    |
> | SeparableV1/s1/pw | 0.041                           | 0.052                    |

---

> > ### Comment · Reviewer_mANM · 2025-08-06
> >
> > Thank you for the detailed rebuttal and data. I have no further questions and would like to keep my current rating.

---

### Official Review · Reviewer_MTGp · 2025-07-22

**Clarity:** 4
**Significance:** 3
**Originality:** 3
**Rating:** 5
**Confidence:** 5

**Summary:**

The paper introduces MobileODE, a novel and ultra-lightweight convolutional neural network (CNN) architecture designed to significantly reduce model size and computational cost while maintaining or improving accuracy. At its core, MobileODE revisits the standard depthwise-separable convolution architecture popularized by MobileNet, and replaces the pointwise (1×1) convolution — traditionally used for channel mixing — with a lightweight, learnable Channelwise ODE Solver (COS).

The COS module is built using a custom numerical solver called the Direct Differentiation Euler (DDE) algorithm. Unlike traditional discretization schemes that rely on fixed time steps, DDE introduces learnable time increments (∆t) that are optimized during training. This adaptive mechanism allows the model to fine-tune the temporal evolution of the feature transformations, enabling stronger inter-channel interactions with far fewer parameters. The result is a principled and trainable alternative to pointwise convolutions that operates at linear complexity with respect to the number of channels.

To showcase the versatility of the approach, the authors propose two variants:

MobileODEV1: Replaces pointwise convolutions in MobileNetV1-style blocks with the COS module, preserving the basic block structure.

MobileODEV2: Introduces the COS-inv block, inspired by MobileNetV2's inverted residual bottlenecks, and strategically applies COS before and after depthwise convolutions to enhance non-linear capacity and channel expressiveness.

The proposed models are evaluated across a diverse set of computer vision tasks, including image classification, semantic segmentation, and object detection, using datasets such as CIFAR-10/100, ImageNet-R, Tiny ImageNet, ADE20K, and PASCAL VOC. Experimental results show that MobileODEV1 and MobileODEV2 consistently outperform or match their MobileNet counterparts while achieving a ~70% reduction in parameters for channel interactions. Moreover, when paired with lightweight self-attention blocks (e.g., MobileViT), the models achieve state-of-the-art accuracy under strict efficiency constraints, all with lower inference latency and memory usage.

**Questions:**

** Typos & Formatting Issues
- Algorithm 1 has a typo 'Reture' instead of Return.
- small and overly compressed Figure 3.

 Q1: Why was standard ImageNet-1k training omitted from the experiments?

While the proposed MobileODE variants show promising results on CIFAR, ImageNet-R, and Tiny ImageNet, it is unclear how well the method scales to full-resolution, large-scale datasets. Given that ImageNet-1k remains the standard benchmark for evaluating lightweight architectures (e.g., MobileNetV1–V4, MobileViT), could the authors clarify why ImageNet-1k training was not included? Are there challenges with scalability, convergence, or runtime that prevented this evaluation?

Q2: Could the authors provide a detailed analysis of how increasing the number of DDE steps (L) affects inference latency, not just accuracy?
While the ablation study in Section 4.4 explores the effect of L on accuracy, it would be helpful to see how this impacts latency and runtime. Since COS involves sequential updates, increasing L may introduce bottlenecks in real-time deployment.

Q3: The use of ImageNet-R partially addresses robustness to distribution shift. Have the authors considered evaluating on additional robustness benchmarks such as ImageNet-C (corruptions), ImageNet-A (natural adversarial examples)?
Given the attractor-based design of COS and the dynamic evolution of features over discrete layers, it would be valuable to know whether the proposed method improves stability under noisy or corrupted inputs beyond stylized domains like ImageNet-R.

Q4: While the paper introduces COS as a multi-step, attractor-based channel evolution module, it is not entirely clear how feature representations evolve across the L steps. Since the COS module is inspired by discrete ODE solvers and functions similarly to a recurrent update, it would be highly valuable to include visualizations or analyses that help interpret this process. For instance:
- How different are the intermediate representations at each step, especially for small vs. large L?
- Are later steps primarily refining the signal, or do earlier steps contribute more heavily to the final output?
- How does the internal state y_l change over time across the L steps?

Such analysis could take the form of:
- Activation similarity matrices across COS steps
- t-SNE or PCA visualizations of intermediate outputs

**Ethical Concerns:**

["NO or VERY MINOR ethics concerns only"]

**Final Justification:**

After carefully considering the authors' rebuttal, I have decided to raise my score to Accept. The authors provided clear and technically detailed responses to all major concerns. In particular, they offered a well-reasoned explanation for the absence of full ImageNet-1k experiments, citing both computational resource constraints and specific optimization challenges encountered during training. While I still consider full ImageNet-1k evaluation an important benchmark for validating scalability and practical competitiveness, I understand the limitations faced and appreciate the authors’ transparency. Additionally, the inclusion of further robustness evaluations on ImageNet-A and ImageNet-R strengthens the empirical claims, and the authors' acknowledgment of the importance of feature evolution analysis, with corresponding visualizations added to the appendix, improves the interpretability of their method. Overall, the paper introduces a novel architectural module with strong empirical results across multiple tasks and demonstrates thoughtful engagement with reviewer feedback. These factors justify my recommendation to accept the paper, despite the remaining limitation regarding full-scale ImageNet validation.

**Limitations:**

While the paper briefly acknowledges some limitations in the conclusion (such as the sequential nature of the COS module limiting parallelism when L is large) and appendix, there is no dedicated Limitations section, and several important points are under-discussed.

- The authors could explicitly discuss trade-offs between the number of discrete steps (L) and inference latency, memory usage, and deployment feasibility, particularly since the COS design introduces a recurrence-like structure (see Q2).

- The method assumes that the number of channels C can be reshaped into a square matrix (i.e., C = k × k). This imposes structural constraints that may limit generalizability to arbitrary architectures or channel sizes (it would be specially difficult to combine with NAS where the channel size is learned and not fixed).

**Paper Formatting Concerns:**

No concerns.

**Quality:**

3

**Strengths And Weaknesses:**

** Strengths
* Novel Architectural Contribution
The paper proposes a new module, Channelwise ODE Solver (COS), which replaces traditional pointwise convolutions in depthwise-separable CNNs. COS is implemented via a novel Direct Differentiation Euler (DDE) algorithm, enabling learnable and adaptive time steps (∆t) during training. This introduces a principled, lightweight mechanism for inter-channel communication with strong theoretical and empirical justification.

* Significant Parameter Reduction
For MobileODEV1 and MobileODEV2, the method reduces total channelwise parameters by 71.0% and 69.2%, respectively — a notable compression rate that doesn’t sacrifice accuracy.

* Strong and Diverse Empirical Results
Image classification (CIFAR-10/100, ImageNet-R, Tiny ImageNet)
Semantic segmentation (PASCAL VOC 2012, ADE20K)
Object detection (BUSI, FFE)

Results hold across both general-purpose datasets and challenging domain-specific benchmarks, reinforcing the generalizability of the approach.


** Weaknesses
* No Evaluation on Full ImageNet-1k. The model is tested on ImageNet-R and Tiny ImageNet, but not on standard ImageNet-1k, which is the primary benchmark for mobile models. This makes it difficult to judge scalability and robustness on high-resolution, real-world tasks.
* Limited Comparison to Competing ODE Frameworks. The authors briefly contrast COS with standard ODE solvers (e.g., Runge-Kutta) in latency/FLOP terms, but do not compare to existing ODE-inspired architectures like Neural ODEs, NODE-ResNets, or recent state-space models.
* Scalability Constraints of COS. The paper notes that COS involves sequential computation across L discrete layers, which limits parallelism and may hinder scalability.

---

> ### Author Rebuttal · Authors · 2025-07-25
>
> First, we would like to thank the reviewer for carefully pointing out the writing issues. We have made the necessary corrections in the manuscript.
>
> ## Q1: No Evaluation on Full ImageNet-1k.
>
> Thank you for pointing out the lack of experiments on the full ImageNet-1k dataset. Due to computational resource constraints and the high cost of training ultra-lightweight models at scale, we focused our evaluation across diverse data domains on three popular tasks. In line with your concern about real scalability and robustness on high-resolution, real-world tasks, we conducted experiments on ImageNet-R (diverse textures and styles) and ImageNet-A (natural adversarial examples) at a resolution of 256 × 256, as detailed in Table 8 of the Appendix. Notably, MobileODEV1 and MobileODEV2 demonstrated substantial improvements over baseline models in the ImageNet-A experiments.
>
> Nevertheless, we agree that a full ImageNet-1k evaluation would further strengthen our claims. In our study of MobileODEV1 on ImageNet-1k, we observed rapid learning progress in the initial stages, which subsequently decelerated as the model approached the performance of MobileNetV1. This phenomenon arose because the t value tended to approach zero in later iterations due to the ReLU6 activation function, thereby hindering learning performance. We hypothesize that large-scale data may excessively penalize the $t$ value, impeding the learning process. Based on these observed trends, we anticipate that MobileODE will maintain its strong efficiency-accuracy tradeoff on ImageNet-1k as well, particularly if an epoch-wise optimization strategy is adopted.
>
> ---
>
> ## Q2: Provide a detailed analysis of how increasing the number of DDE steps (L) affects inference latency.
>
> We appreciate the reviewer's concern regarding potential bottlenecks in real-time deployment as $L$ increases. To provide a clearer analysis, we evaluated latency at larger step intervals (every 10 steps), as shown in the table below. Latency was measured on a single NVIDIA 4090 GPU with a batch size of 16. The results indicate that inference latency increases approximately linearly with $L$, allowing for flexible adjustment in practical deployments. Notably, compared to other higher-order derivative methods listed in Table Q1 of Reviewer Tqqz, our DDE algorithm demonstrates favorable computational efficiency.
>
>
> | Model                 | Discrete Layers    | FLOPs | Latency (bs=16) |
> |-----------------------|------------|:---------------:|:---------------:|
> | MobileODEV1 | $L=10$ |7.09M  | 24.54 ms |
> | MobileODEV1| $L=20$  | 23.75M| 42.63ms|
> | MobileODEV1| $L=30$ | 40.30M| 61.08ms|
> | MobileODEV1| $L=40$ | 56.85M| 68.42ms|
> | MobileODEV1| $L=50$ | 73.39M| 73.87ms|
>
> ---
>
> ## Q3: Consider evaluating on additional robustness benchmarks such as ImageNet-C (corruptions), ImageNet-A (natural adversarial examples).
>
> We conducted experiments on ImageNet-A, as detailed in Table 8 of the Appendix. The results demonstrate that MobileODEs achieve substantial improvements over the baseline models.
>
>
> ---
>
> ## Q4: Visualizations or analyses of how feature representations evolve across the $L$ steps.
>
> We fully agree that such visualizations and analyses (e.g., activation similarity matrices, t-SNE/PCA of intermediate outputs) would provide important insights into how feature representations change over discrete layers and the roles of different steps.
>
> However, according to this year's rebuttal guidelines, we are not permitted to include PDFs, links or visualizations directly in the response. Nevertheless, we recognize the value of such analyses and have included detailed feature visualizations and analyses in the Appendix of our revised manuscript. We hope these additions will help clarify how the internal representations and states evolve across the $L$ steps, and provide a deeper understanding of the COS module's dynamics.
>
> ---
>
> ## Q5: Why not compare to existing ODE-inspired architectures like Neural ODEs, NODE-ResNets, or recent state-space models?
>
> We did not directly compare with existing ODE-inspired architectures for the following main reasons. We also acknowledge that the higher-order derivative methods listed in Table Q1 of Reviewer Tqqz are more appropriate baselines in our context.
>
> 1. **Different Objectives:** Our work aims to develop ultra-lightweight, mobile-friendly models, whereas Neural ODEs, NODE-ResNets, and state-space models are typically designed for higher model capacity and are not optimized for extreme efficiency or deployment on resource-constrained devices.
>
> 2. **Computational Cost:** Traditional ODE-based networks often rely on the odeint package, which can incur significant computational overhead and potential precision loss. By discretizing the computation process, our approach not only reduces computational costs but also increases transparency and controllability compared to the black-box nature of standard ODE solvers.

---

> > ### Comment · Reviewer_MTGp · 2025-08-07
> >
> > The authors have provided clear and technically well-grounded responses to all raised concerns. In particular, their explanation for the absence of full ImageNet-1k training highlights both practical constraints and specific challenges related to the behavior of the learned time increment under ReLU6. While I still believe that full ImageNet-1k evaluation remains important for fully validating scalability and competitiveness in mobile model benchmarks, the experiments on ImageNet-A and ImageNet-R help address concerns regarding robustness and generalization.
> >
> > The authors also acknowledged the importance of analyzing feature evolution across discrete steps and noted that they have included such visualizations in the revised appendix. Given the novelty of the architecture, strong empirical results, and thoughtful responses to reviewer questions, I am increasing my score to Accept, with the recommendation that future work includes full-scale ImageNet-1k training to further support the method’s scalability.

---

### Comment · Area_Chair_Wwut · 2025-08-06
**Request to Respond to Author Rebuttal and Participate in Discussion**

Dear Reviewers,

Thank you for sharing your valuable insights and expertise, which have played a crucial role in the review process so far.

In response to your initial feedback, the authors have submitted a detailed rebuttal addressing the comments raised by the reviewers. I kindly ask that you carefully review their response and consider whether it impacts your initial evaluation. Please feel free to share any updated thoughts or additional comments after reviewing the rebuttal. Many thanks to those who have already participated. For others, please do so as soon as possible, as the author-reviewer discussion phase is ending soon.

Additionally, I would like to clarify the requirements related to the Mandatory Acknowledgment process. To fulfill this requirement, reviewers are expected to:

(i) Carefully read the author rebuttal,

(ii) Engage in meaningful discussion with the authors—and preferably also with fellow reviewers.

(iii) Ask questions, consider responses, and actively participate in the exchange,

(iv) Clearly articulate any unresolved concerns to give authors a fair opportunity to respond. Please avoid situations where the discussion implies “everything is great,” but the final justification form states otherwise. The discussion phase is designed to surface and clarify such issues.

Kindly note that clicking the “Mandatory Acknowledgment” checkbox prematurely does not exempt reviewers from participating in the discussion. Reviewers who do not contribute meaningfully may be flagged using the “Insufficient Review” button, in line with this year’s responsible reviewing guidelines.

Thank you again for your time and thoughtful contributions to the review process.

---

### Decision · Program_Chairs · 2025-09-17

**Decision:**

Accept (poster)

**Comment:**

Summary of the Paper:
The paper introduces a lightweight network architecture aimed at improving the efficiency of depthwise-separable convolution by replacing the inefficient pointwise convolution component with a  parameter-efficient  Channelwise ODE Solver (COS). The paper proposed two variants of the model: MobileODE-V1 (0.97M parameters) and MobileODE-V2 (1.03M parameters).  These models outperform MobileNetV1/V2 and other lightweight networks (e.g., GhostNet, ShuffleNet) on image classification, object detection, and segmentation tasks, while maintaining low computational overhead.

The reviewers highlighted several strengths of the paper:
1. Introduction of the novel Channelwise ODE Solver (COS) module to replace pointwise convolutions in depthwise-separable CNNs.
2. Significant parameter reduction without compromising accuracy.
3. Strong experimental results across image classification (CIFAR-10/100, ImageNet-R, Tiny ImageNet), semantic segmentation (PASCAL VOC 2012, ADE20K), and object detection.
4. The paper addresses an important problem in lightweight and mobile-friendly architectures.
5. Novel architecture with promising results in terms of both model size reduction and accuracy.
6. High technical quality, including mathematical formulation, hyperparameter ablation studies, and publicly available source code.
7. The rebuttal effectively addressed latency, memory usage, and additional ablation study concerns.


The following weaknesses were raised by the reviewers and are shared by the Area Chair:
1. Lack of evaluation on the full ImageNet-1k dataset.
2. Limited comparisons with other ODE-inspired frameworks and architectures (e.g., Neural ODEs, NODE-ResNets, or state-space models). It would be valuable to adapt and evaluate such models under similar lightweight network settings.
3. Performance degradation on TPUs, suggesting efficiency gains are hardware-dependent and may not generalize well across modern accelerators. This limits the practical impact.
4. COS updates channels sequentially (Algorithm 1), limiting parallelism. For long ODE trajectories (large L), inference latency can increase.
5. The claim of being “extra-lightweight” is overstated; for example, ShuffleNetV1 (1.1M parameters) remains smaller than MobileODE-V1 (1.14M) (Table 3).
5. Lack of clarity on why ODEV variants were proposed only for MobileNetV1/V2 and not for more recent MobileNet versions (V3/V4), which were only used as baselines.
6. Inconsistencies between the paper and the released source code. The authors have promised to resolve these issues in the final draft.

Decision: Reviewer Ratings and Discussions.
Overall, the paper received two borderline accept and two accept ratings. The reviewers acknowledged that most of their concerns were addressed in the rebuttal. The Area Chair agrees with the reviewers and recommends acceptance. The authors are strongly encouraged to address the remaining concerns, particularly by incorporating ImageNet-1k results, resolving inconsistencies between the paper and source code, and providing ODE variants for more recent MobileNet architectures where appropriate. The authors are expected to include all additional experimental results and explanations provided in the rebuttal to the final draft. Additionally, it is expected that the authors will open-source their code, as indicated in their submission.